# Importations of SARS-CoV-2 lineages decline after nonpharmaceutical interventions in phylogeographic analyses

Sama Goliaei[1,2], Mohammad-Hadi Foroughmand-Araabi[1,2], Aideen Roddy[1,2], Ariane Weber[3], Sanni Översti[3], Denise Kühnert[3,4,5] & Alice C. McHardy ⓘ [1,2,4] ✉

During the early stages of the SARS-CoV-2 pandemic, before vaccines were available, nonpharmaceutical interventions (NPIs) such as reducing contacts or antigenic testing were used to control viral spread. Quantifying their success is therefore key for future pandemic preparedness. Using 1.8 million SARS-CoV-2 genomes from systematic surveillance, we study viral lineage importations into Germany for the third pandemic wave from late 2020 to early 2021, using large-scale Bayesian phylogenetic and phylogeographic analysis with a longitudinal assessment of lineage importation dynamics over multiple sampling strategies. All major nationwide NPIs were followed by fewer importations, with the strongest decreases seen for free rapid tests, the strengthening of regulations on mask-wearing in public transport and stores, as well as on internal movements and gatherings. Most SARS-CoV-2 lineages first appeared in the three most populous states with most cases, and spread from there within the country. Importations rose before and peaked shortly after the Christmas holidays. The substantial effects of free rapid tests and obligatory medical/surgical mask-wearing suggests these as key for pandemic preparedness, given their relatively few negative socioeconomic effects. The approach relates environmental factors at the host population level to viral lineage dissemination, facilitating similar analyses of rapidly evolving pathogens in the future.

The Severe Acute Respiratory Syndrome Coronavirus 2 (SARS-CoV-2) is a rapidly spreading, highly infectious virus of zoonotic origin causing Coronavirus Disease 2019 (COVID-19)[1]. SARS-CoV-2 was first identified in December 2019 in Wuhan, China, and continued to spread globally, resulting in approximately 770 million confirmed infections and seven million deaths to date[2,3]. Prior to the widespread availability of vaccines, nonpharmaceutical interventions (NPIs) such as reducing contacts, antigenic testing, and travel restrictions were the primary means of reducing SARS-CoV-2 transmission and case numbers in the pandemic.

To trace viral spread across countries, as well as the emergence of novel variants with concerning phenotypic alterations, viral genomic surveillance was established in many countries. This resulted in unprecedented amounts of genome sequences being rapidly generated worldwide. By combining genomic information with sampling times and locations, the spatial dispersal and lineage evolution can be reconstructed with viral phylogeographic techniques. Phylogeographic analyses provide insight into SARS-CoV-2 importations into Europe at the beginning of the pandemic[4]. For example, studies

---

[1]Computational Biology of Infection Research, Helmholtz Centre for Infection Research, Braunschweig, Germany. [2]Braunschweig Integrated Centre of Systems Biology (BRICS), Technische Universität Braunschweig, Braunschweig, Germany. [3]Transmission, Infection, Diversification and Evolution Group, Max-Planck Institute of Geoanthropology, Jena, Germany. [4]German COVID Omics Initiative (deCOI), Bonn, Germany. [5]Centre for Artificial Intelligence in Public Health Research, Robert Koch Institute, Wildau, Germany. ✉e-mail: Alice.McHardy@helmholtz-hzi.de

suggest that it was transported from Hubei, China, to multiple European countries several times between mid-January and early February 2020, before the large outbreak in northern Italy. The first wave of infections was studied in the United Kingdom (UK, late winter and early spring 2020) and Portugal (fall 2020), both of which had high early sequencing rates and therefore allowed to characterize the importation and diversity of lineages in depth[5,6]. For the UK, international travel led to the importation of over 1000 co-circulating transmission lineages. Both for the UK and Portugal, most introductions occurred prior to lockdown measures (UK lockdown on 2020-03-23, involving the closure of non-essential shops and services, and a stay-at-home order; Portugal lockdown on 2020-04-09 restricting people's movements between municipalities, closing air travel, and hardening border control), with the earliest ones becoming the largest and most persistent lineages post-lockdown[5]. Although Portugal quickly implemented lockdown measures, SARS-CoV-2 was likely circulating in late February, weeks before the first detected case[6]. Contrasting this, a study of Belarus reported few early importations that were largely brought in from neighboring countries, which is consistent with travel data[7].

Following the lifting of the first lockdown measures (Jun 2020 in UK), the B.1.117 lineage spread in a second infection wave across Europe over the summer of 2020[8,9], leading to a persistently high volume of cases, despite B.1.117 having no notable transmission advantage. Further, a study of the third and fourth infection waves in Hong Kong (Jul and Nov 2021) provides insight into the successes and challenges of an elimination strategy[10], as opposed to the mitigation approach adopted by many other countries. As a result of strict border control measures, there was a low level of importation with the third and fourth waves, which occurred over the period July 2020–April 2021. Local transmission was much higher in wave four than wave three, which was attributed to reduced compliance with control measures due to pandemic fatigue[10]. Ultimately, SARS-CoV-2 was eliminated during the study period due to contact tracing and mandatory quarantine measures. Further, low levels of importation were maintained due to highly stringent border controls, including a 21-day quarantine on arrival. Similarly, a Bayesian phylogeography analysis showed that Switzerland's strict border closures alongside the 2020 partial lockdown were effective in controlling the entrance of new lineages into the country[11]. However, later VOCs (Alpha, Delta, and Omicron) exhibited increased transmissibility, making an elimination strategy much more challenging to maintain.

The third infection wave in Europe during spring 2021 largely consisted of the B.1.1.7 (Alpha) lineage, which was first detected in Kent or Greater London[12]. Alpha exhibited higher transmissibility compared to previous ones and was not contained by the UK lockdown measures[13], which controlled other lineages in late 2020[14]. Instead, stricter measures had to be introduced in early 2021[15]. Studying both the third and fourth waves in England (spring and summer of 2021), characterized by Alpha and Delta variants, respectively, revealed that their growth was initially masked by falling case counts of more dominant lineages[16].

Systematically assessing the effects of different NPIs on viral lineage importations and spread is key for future pandemic preparedness, although currently, these are not entirely clear[17]. Here we systematically analyzed how the NPIs that were successively implemented within a country affected SARS-CoV-2 lineage importations. For this, we used data from representative genomic surveillance collected over the course of the third pandemic wave in Germany (late 2020 and early 2021), together with large-scale Bayesian phylogenetic analyses. We then identified stable properties across multiple sampling strategies that consistently led to fewer importations into the country and less dissemination across states.

## Results

### SARS-CoV-2 lineage importations into Germany

Nationwide viral surveillance through genome sequencing, covering up to 5–15% of registered cases[18], commenced in Germany in January

2021, before the third pandemic wave gained momentum in February 2021[19]. Of the SARS-CoV-2 isolates sequenced between February and May 2021 in Germany, 71.95% belonged to the B.1.1.7 lineage, with an increasing frequency over time. Additionally, 11.30% belonged to the B.1.177 lineage, which had previously spread across Europe[9]. Notably, B.1.1.7 was predominant among sequenced isolates from the German states of North Rhine-Westphalia, Bavaria, and Baden-Württemberg.

The genomic surveillance program allowed us to conduct a systematic analysis of viral lineages imported into the country during this period using large-scale phylogenetic analysis techniques. We adapted a Bayesian phylogeographic approach to infer a phylogeny from publicly available SARS-CoV-2 genomes up to 2021-06-02[5]. This global dataset consisted of 1.8 million viral genome sequences, with varying coverage of case numbers in different countries and time periods due to differences in sequencing rates. To address geographic and temporal sampling biases, assess the stability of results, and identify consistent properties across different data subsets, we created three distinct datasets using previously employed sampling strategies[7,8,11], namely Case ratio subsampling, 50:50 subsampling, 25:100 subsampling (Methods, Table S1, Data sampling schemes in Supplementary Methods). From each of these datasets, we identified SARS-CoV-2 lineages imported into Germany, along with their estimated importation times (see "Methods" section).

Many properties remained consistent across the three samplings, while some properties varied. With an increasing number of non-German sequences included in the datasets, the age of the TMRCAs decreased, while the number of importation lineages increased (50:50 vs. 25:100; Table S1, Fig. S2B, C). This suggests that the inclusion of more non-German sequences results in the partitioning of samples of an inferred lineage into multiple lineages, thereby resulting in later TMRCAs and smaller importation lineages. Consequently, since our sampling of non-German sequences was more limited than of German sequences, the number of lineages genuinely imported into Germany are likely to be higher than the obtained estimates.

We examined the largest lineages that were imported and their subsequent spread, as these had the most substantial impact on Germany during the third pandemic wave. Consistently, only a few large lineages were detected (Figs. 1A, S2, Table S2). In the main sampling, proportional to the number of cases, the 15 largest importation lineages cover 20.3% of the German samples, while the remaining samples were assigned to 661 additional lineages. Also, the absolute number of identified importation lineages varied across samplings. This aligns with sizes of the importation lineages following a power-law distribution, as seen for the UK[5], where growth is primarily seen for the large lineages, and most lineages are small[20].

Most importation lineages belong to B.1.1.7, the main driver of the third pandemic wave in Germany. Notably, the B.1.1.7 importation lineages responsible for most cases were imported within a two-month period, spanning from mid-November 2020 to the end of December 2020 (Fig. 1B). The overall number of inferred importation lineages began to rise shortly before and reached its peak after the Christmas holidays (Fig. 1C), indicating that holiday-related travels played a key role in importations and the spread of lineages within the country, consistently across samplings.

### Appearance and diversity of importation lineages within the country

Most importation lineages first appeared in North Rhine-Westphalia (25.1%), Bavaria (15.8%), and Baden-Württemberg (7.1%; Fig. 1C) in all samplings, across all Pangolin lineages and B.1.1.7-associated importation lineages only (Fig. S3). This indicates the relevance of these three most populous German states for lineage importation and within-country dissemination during the third wave. Consistently across all sampling methods, the fewest lineages first appeared in Mecklenburg-Western Pomerania and Brandenburg, both 0.5% (Fig. S4). The initial

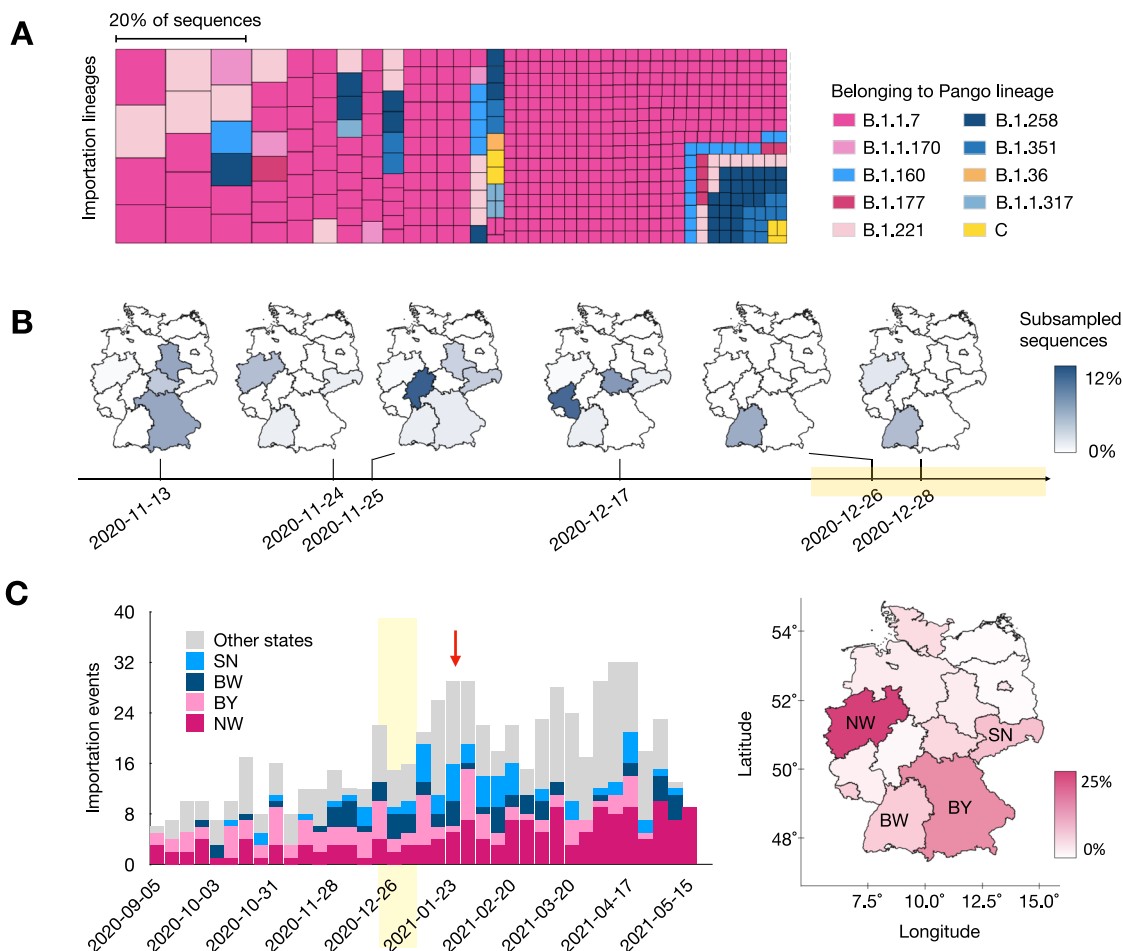

**Fig. 1 | Epidemiological properties of viral importation lineages. A** Importation lineages are colored by the corresponding Pango-lineage to which they belong. The size of squares represents the size of the importation lineages. The figure exclusively displays lineages that were not eliminated before 2021-01-01. **B** Estimated importation time and geographic distribution over states for the six largest B.1.1.7 importation lineages. The colors represent the number of viral genomes belonging to the respective importation lineage for each individual state. The light orange block indicates the time of the Christmas holidays (2020-12-24 to 2021-01-09). The holidays vary by state, starting on 2020-12-24, and ending between 2021-01-02 and 2021-01-09. **C** Number of importation lineages entered into the country per week,

colored by the state that they were first observed in, shown individually for the states with the most importation events (dark pink: North Rhine-Westphalia, pink: Bavaria, dark blue: Baden-Württemberg, blue: Saxony) and for the remaining states in gray. The light orange block denotes the Christmas holidays, from 2020-12-24 to 2021-01-02 until 2021-01-09, depending on the state, and the red arrow indicates the peak in importation events after the Christmas holidays. See Fig. S1 for the results for other samplings. The map represents the percentage of detected importation lineages, first observed in individual states (white: fewer, pink: more). NW = North Rhine-Westphalia, BW = Baden-Württemberg, BY = Bavaria, SN = Saxony.

appearances correlate with the states' population sizes (Pearson Correlation Coefficient (CC) = 0.82) and confirmed case numbers (Pearson CC = 0.86). Although sequencing rates varied for states (Pearson CC of sequencing rate and population = −0.31, Fig. S5), there was no correlation with state-wise sequencing rates of the representative SARS-CoV-2 sequencing program initiated for Germany in January 2021 (Pearson CC = −0.01, Table S3, Fig. S6).

Notably, despite having the airport with the most annual number of travelers in Germany, comparatively few (<2%) importation lineages were first observed in Hesse. Among the ten most frequented airports in Germany in 2020, three are located in the state of North Rhine-Westphalia (ranked third, eighth, and tenth), one in Bavaria (ranked second), one in Baden-Württemberg (ranked seventh), while none is found in Mecklenburg-Vorpommern and Brandenburg[21]. While we anticipated that air travel would have a substantial impact on lineage importations, we observed that the relationship to the state-wise population was more pronounced, indicating that the virus is more likely transmitted at their final destination. We hypothesize that the state population is more reflective of the ultimate destination of these

travels, as large airports often serve only as intermediate stops for travelers.

To assess the diversity of importation lineages within states, we calculated the Shannon Index (SI), as by Spellerberg et al.[22], and the importation lineage evenness for each state. The SI is influenced by the total number of imported lineages and their respective size distributions, where having many imported lineages of similar sizes results in larger SI values. Across samplings, the state-wise SI strongly correlates with the logarithm of the number of analyzed sequences. In the case of evenness, the SI of a state is divided by the number of imported lineages present in that state[23], providing a direct reflection of the importation lineage size distribution. SIs were highest for the population-rich states of North Rhine-Westphalia, Bavaria, and Baden-Württemberg, and Saxony, as expected due to the large number of cases and importation lineages observed. The lowest SI was determined for Schleswig-Holstein (Fig. 2A), which exhibited a substantially lower SI value than several other states with fewer confirmed cases and circulated importation lineages, as the lineage size distribution is strongly skewed towards one dominant lineage that entered the state

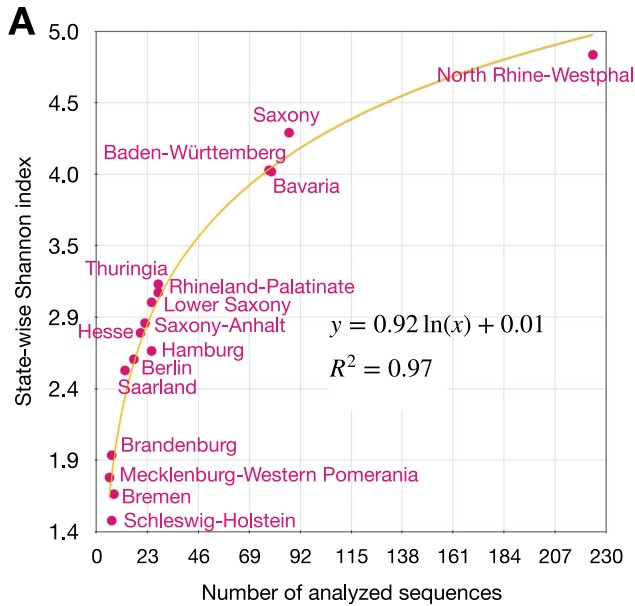

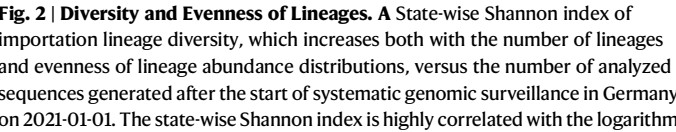

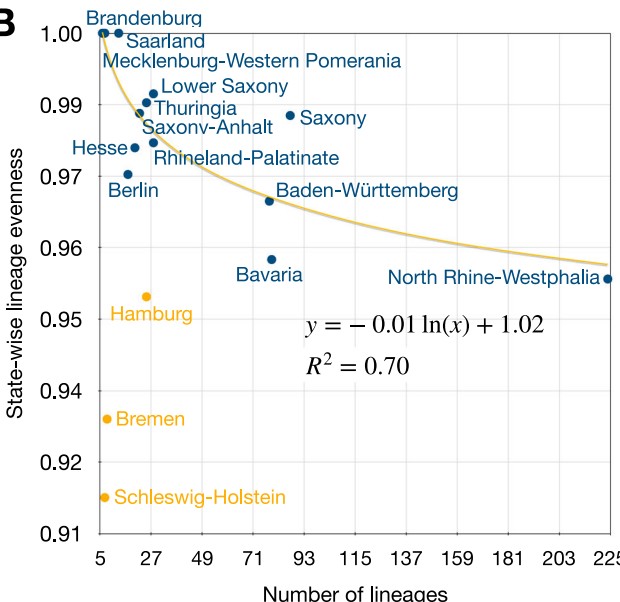

**Fig. 2 | Diversity and Evenness of Lineages. A** State-wise Shannon index of importation lineage diversity, which increases both with the number of lineages and evenness of lineage abundance distributions, versus the number of analyzed sequences generated after the start of systematic genomic surveillance in Germany on 2021-01-01. The state-wise Shannon index is highly correlated with the logarithm of the number of analyzed sequences. **B** State-wise evenness of lineage abundance distributions versus the number of analyzed sequences generated after 2021-01-01. The orange curves (**A**, **B**) indicate a fitted logarithmic function. R2 denotes the extent of variance explained by the fitted functions, with orange points indicating outliers excluded from the analysis.

before the Christmas holidays (Fig. S7), indicating exceptionally low connectivity to other regions. This result remains stable across samplings (Fig. S8). With the exception of Hamburg, Bremen, and Schleswig-Holstein, lower lineage numbers correlated with more equal lineage sizes, higher evenness, across states (Fig. 2B, S8). While these relationships are not unambiguously resolved[23], such effects can be explained by niche preemption, which posits that a highly diverse environment, i.e., one with many lineages, is more challenging for incoming lineages to invade.

## Lineage importations are reduced after nonpharmaceutical interventions

Throughout the study period, multiple NPIs were implemented and adapted both at the federal and state levels to control viral transmission and case numbers in the population, including both nationwide NPIs and state-specific ones. To evaluate how NPIs affected the rate of lineage importations into the country and states, we summarized information of more than 4,000 NPIs implemented in Germany from published sources[24–27] into 119 NPIs (Table S4), and then categorized the national and local NPIs into 12 major NPIs (Fig. 3), and then classified them as internal NPIs (N1[gathering restriction], N2[partial lockdown], N4[contact restriction], N5[lockdown], N8[mask], N11[free test]), border control NPIs (N3[travelers registration], N6[UK travelers test], N9[variant travelers ban], N10[border closure], N12[air travelers test]), and NPI N7[15 km movement ban], which includes both internal and border control measures. Internal NPIs include measures for school closures, workplace closures, canceling public events, restrictions on gatherings, restrictions on people's movements, obligation to wear surgical/FFP2 masks in public transport (from wearing of any kind of mask, including homemade cloth masks, to medical FFP2 or surgical masks), and availability of free antigen tests. Among the border control NPIs, N3[travelers registration], N6[UK travelers test], and N12[air travelers test] made regulations more restrictive until 2021-05-09, when these NPIs were relaxed by allowing vaccinated people to bypass the mandatory quarantine upon arrival. Regarding internal measures, the six NPIs (N1[gathering restriction], N2[partial lockdown], N4[contact restriction], N5[lockdown], N7[15 km movement ban], and

N8[mask]) made measures stricter until mid-February 2021 when these restrictions were gradually relaxed.

We defined a daily effectiveness measure (DEM) based on the regulations put into effect, from the number of importation events in the previous 7 and the next 14 days (Methods). Lineage importations into the country were reduced after all twelve major nationwide NPIs as measured by this effectiveness (Fig. 4). Across samplings, the availability of free rapid tests starting from 2021-03-08 (N11[free test]) was the most effective NPI for the country (DEM = 31), along with the strengthening of the mask-wearing in public transports and shops on 2021-01-24 (N8[mask]), which made surgical/FFP2 mask-waring mandatory instead of wearing any kind of mask, which was required before this. Additionally, the restrictions implemented on 2021-01-11 (N7[15 km movement ban]), which include limiting movements to a 15 km radius, restriction of gatherings to be with at most one person from another household, and mandatory testing for all travelers from risk areas, demonstrated high effectiveness in two sampling strategies (Fig. S9).

While disentangling the separate factors affecting importations is a challenging task, several factors enhance confidence in attributing the observed reduction in importations mainly to the implementation of NPI N8[mask] on January 24 rather than solely to the conclusion of the Christmas holidays. To understand the extent to which the ending Christmas holidays' have an effect on importations, we examined the importation pattern in the three most populous states. We exclude Bavaria from this observation, since Bavaria first adopted the internal 15 km movement ban on 2021-01-11, then relaxed it a few days later. Examining lineage importations in Baden-Württemberg, a highly populated state that did not implement the internal movement ban in NPI N7[15 km movement ban], allowed us to analyze the effect of ending Christmas holiday on importations with less NPI influence. The change rate (derivative) of SIF began to increase on 2021-01-09, marking the end of the Christmas school holidays, and decreased on 2021-01-16, occurring nine days before the implementation of N8[mask] on 2021-01-25 (Fig. S10). The same pattern is observable for North Rhine-Westphalia, where the change rate of SIF began to increase on 2021-01-09, and decreased on 2021-01-15, which is ten days

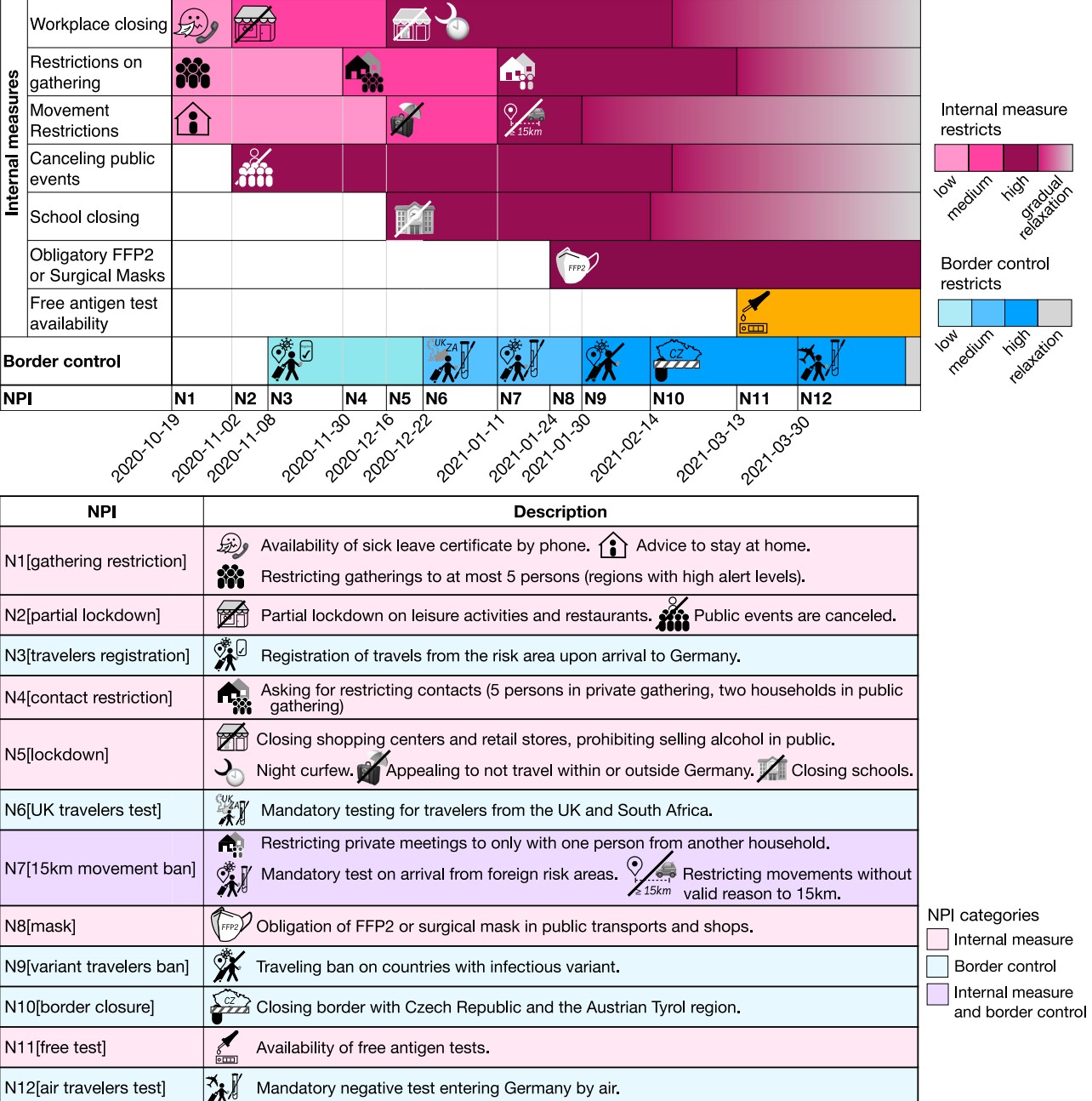

**Fig. 3 | Application date and descriptions for twelve major NPIs implemented in Germany between October 2020 and March 2021.** The first seven rows show the details of internal measures over time, while the next row represents the border control measure. In the pink and blue rows, lighter colors represent less restrictive measures, and darker colors represent more restrictive interventions. Gray color indicates the relaxation of the restrictions applied at the end of the third wave. Note that the presented intensity and colors are for illustrative purposes only. N1[gathering restriction], N2[partial lockdown], N4[contact restriction], N5[lockdown], N8[mask], and N12[air travelers test] are internal NPIs, while N3[travelers registration], N6[UK travelers test], N9[variant travelers ban], N10[border closure], and N12[air travelers test] are border control NPIs. N7[15 km movement ban] includes both internal and border control measures. In the table, the NPIs are described. The pink rows represent internal NPIs, blue rows represent border control NPIs, and the purple row represents the NPI, which includes both internal and border control measures.

before the implementation of N8[mask] in North Rhine-Westphalia on 2021-01-25. Consequently, we deduce that holiday travels had already decreased before implementation of N8[mask].

When examining the effectiveness of NPIs within individual states, N11[free test] emerged as the most effective NPI for North Rhine-Westphalia (DEM = 9, in two subsampling strategies, case ratio and 50:50), Bremen (DEM = 2, in two subsampling strategies, case ratio and 50:50), and Rhineland-Palatinate (DEM = 6, in all three subsampling strategies, Fig. S11). NPI N8[mask] was the second most effective NPI

for the country (DEM = 24), as well as for Baden-Württemberg (DEM = 9, in all three subsamplings), and Bavaria (DEM = 15, in two subsampling strategies, case ratio and 50:50). Furthermore, NPI (N7[15 km movement ban]) was also the most effective NPI for Lower Saxony (DEM = 6, in all three subsampling strategies) and Bremen (DEM = 2, in two subsampling strategies, case ratio and 50:50).

Overall, internal NPIs (average DEM = 18.83) appeared more effective for the country than border control ones (average DEM = 10). Hamburg, Bremen, Lower Saxony (only in 25:100 and 50:50

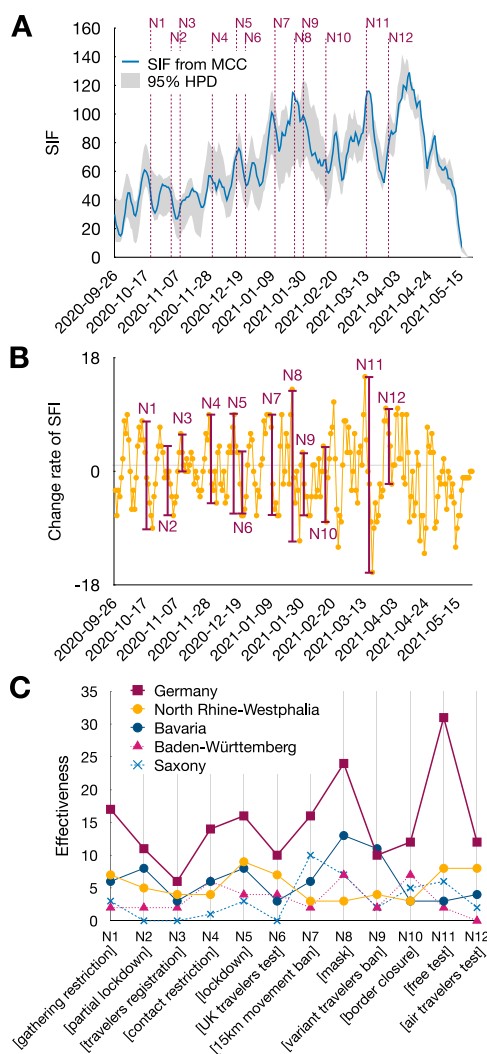

**Fig. 4 | Effectiveness of NPIs. A** Smoothed importation frequency (SIF) values over time. The gray area represents 95% HPD from 2000 Bayesian posterior samples, while the blue line depicts the SIF values derived from the MCC tree. **B** Change rate of SIF over time calculated from the MCC tree. Red vertical bars indicate the implementation date of the twelve major NPIs in the country, and their heights represent the effectiveness metric for that day. **C** NPI effectiveness for Germany overall, as well as for North Rhine-Westphalia, Bavaria, Baden-Württemberg, and Saxony.

subsamplings), Hesse, Saarland, Brandenburg, and Berlin (only in the second and third subsamplings) have a higher average of effectiveness in border control NPIs than internal NPIs, indicating the benefit of border control NPIs in these states. This is a stable result across the three sampling strategies (Fig. S11).

Analysis of the sensitivity of the SIF and effectiveness measures show the robustness of the results with respect to minor changes in the smoothing parameter (Supplementary Materials, Sensitivity analysis of the smoothed importation frequency and effectiveness measures, Fig. S12). Furthermore, N11[free test] and N8[mask] are consistently two most effective NPIs for changes in importation lag, from importation lag be set from 1 to 6 days (Fig. S13, Table S5).

The 95% Highest Posterior Density (HPD) for the SIF and effectiveness were analyzed based on samples from the posterior distribution of the two Bayesian steps (Fig. 4A, S14). When considering effectiveness based on the average values (Table S6), N11[free test] exhibited the highest average effectiveness, similar to the MCC tree. The second-highest average effectiveness was observed for N7[15 km

movement ban], which closely resembled N8[mask] (13 days), one of the three most effective NPIs based on the MCC tree.

Owing to effective NPIs, case numbers declined from November 2020 onwards, reaching a minimum in early February of 2021 (Figs. S15–S17). However, this was followed again by a steep rise over the next two months. Already in January, most imported lineages were from the substantially more transmissible B.1.1.7 variant[28], as B.1.1.7 lineages propagated within the country and gained momentum, demonstrating reduced effectiveness of implemented NPIs for this more transmissible lineage or reduced effectiveness of remaining NPIs after relaxation of NPIs in February. This is consistent with the fact that B.1.1.7 was first detected in the UK in September 2020 and became the dominant lineage after a few months (in January it grew rapidly and continued to grow for three more months)[13,28].

### Inter-state lineage spread
We analyzed the spread of lineages within Germany across states using a Bayesian 16-state DTA model and determined the number lineage transfers between states (Methods, Fig. 5A, Table S7). The most lineages (ten) spread from North Rhine-Westphalia to Baden-Württemberg with all subsampling strategies, followed by lineage transmissions from North Rhine-Westphalia to Saxony, from Bavaria to Baden-Würtemberg, and from North Rhine-Westphalia to Lower Saxony. Four states, namely North Rhine-Westphalia, Bavaria, Saxony, and Baden-Württemberg, which except for Saxony are the most highly populated states, played the most substantial role in lineage spread within Germany. Specifically, North Rhine-Westphalia had the most outward-going lineage spread (56), followed by either Bavaria (23) or Saxony in all samplings, while Lower Saxony and Mecklenburg-Vorpommern had no lineage transmissions to any other state (Fig. 5B).

The inter-state spread of B.1.1.7 lineages started to rise before Christmas and reached a peak after the Christmas holidays (Fig. 5B), suggesting the impact of these holidays on the dissemination of these lineages within the country. Another peak occurred in late February, coinciding with the relaxation of internal restrictions, such as the closure of workplaces, movement restrictions to 15 km, and the closure of schools. This suggests that the restrictions imposed by these internal NPIs had previously limited the spread of lineages across states. These two results were consistent in all three subsampling strategies (Figs. S18, S19).

The analysis of the posterior distribution of the 16-state DTA (explained in the section Analyzing effectiveness based on posterior samples of the Bayesian method of Supplementary Materials) shows that the results obtained from the MCC of the 16-state DTA and the average of the number of inter-state transmissions obtained from the Bayesian posterior distribution are similar. Both results fall within the 95% highest posterior density (HPD) region (Fig. S14F).

Notably, the steady decline of importations and inter-state movements evident at the very end of the study period is linked to the end of our study period, as the dataset we used includes data available until June 2nd, 2021 in GISAID (Fig. S20).

## Discussion
The onset of the SARS-CoV-2 pandemic brought about considerable difficulties for both society and healthcare systems. Before vaccines were widely accessible, strategies to control SARS-CoV-2 transmission and cases mainly relied on nonpharmaceutical interventions such as reducing social interactions, using antigenic testing, and imposing travel limits. Studies have shown declining lineage importations and persistence after lockdowns for the UK (commencing on 2020-03-23, involving the closure of non-essential shops and services, and a stay-at-home order) and Switzerland (from 2020-03-17 to 2020-04-27, entailing the closure of bars, shops, and other gathering places except for essential services)[5,11], and decreasing case numbers after establishment of NPIs[29–32]. To prepare for future pandemics and to identify

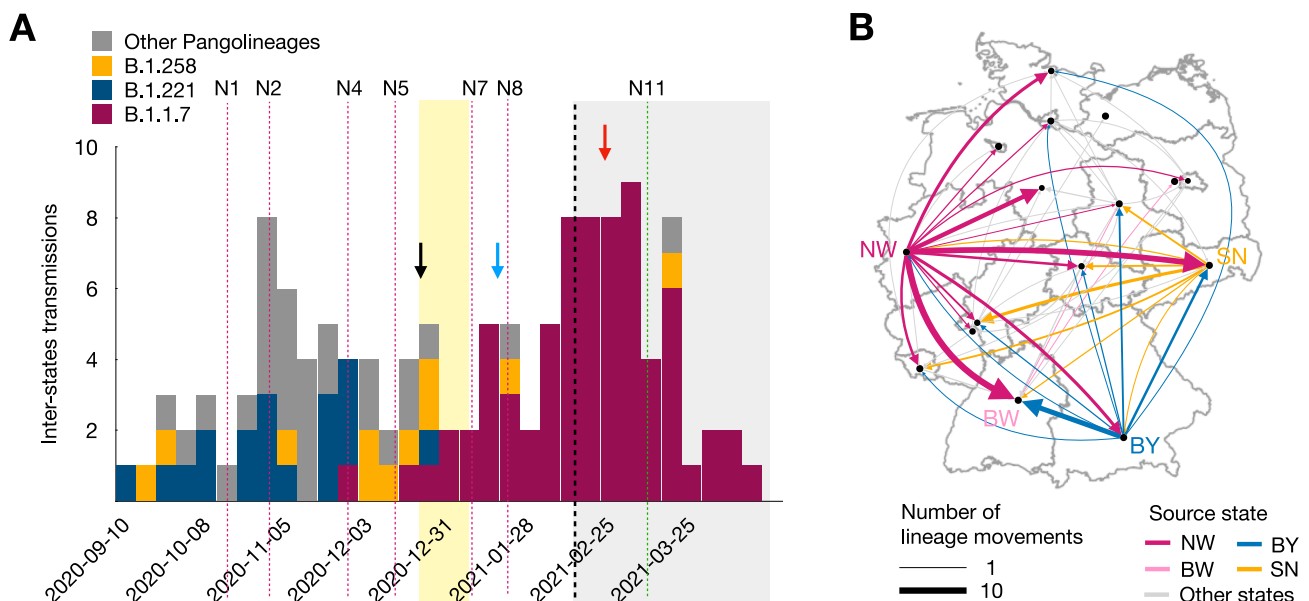

**Fig. 5 | Inter-state spread of lineages. A** Number of inter-state spread of lineages, colored by the Pango-lineage. The number of B.1.1.7 lineage transmissions rise at the beginning of the Christmas holidays and after that (black arrow and blue arrow). There is a peak of B.1.1.7 lineages inter-state transmission in mid-February 2021 (red arrow) when NPI restrictions were gradually relaxed. **B** Spread of transmission lineages across the states within Germany. Arrow widths depict the number of detected migration movements and the color reflects the state of origin. The abbreviations of the states are provided in the legend of Fig. 1.

the most effective measures, it is crucial to evaluate how SARS-CoV-2 lineage importations into the country and their subsequent internal dissemination relate to the nonpharmaceutical interventions that were put into effect. Here, we performed a comprehensive analysis of how SARS-CoV-2 lineage importations were reduced after a range of NPIs that were implemented within Germany over the course of the third infection wave of the pandemic. We assessed viral genomes collected in representative genomic surveillance together with information on their sampling times and locations using large-scale Bayesian phylogenetic analyses, adapting the framework developed by du Plessis et al.[5].

The largest reductions of SARS-CoV-2 lineage importations were seen in Germany after the provision of free rapid tests, and the strengthening of regulations on mask-wearing from wearing any kind of mask to obligatory surgical/FFP2 mask-wearing in public transport and in stores, and on internal movements and gatherings. The provision of free rapid tests and the strengthening of regulations on mask-wearing have substantially fewer socioeconomic effects than restrictions on public gatherings and internal movements, which more strongly affect day-to-day activities[33].

We furthermore determined a particular co-incidence of the German Christmas holidays and highly populated states for SARS-CoV-2 lineage importations into the country and their subsequent internal dissemination, indicating a benefit in considering holiday-related dynamics when planning NPIs. Interestingly, studies of the dissemination of seasonal influenza viruses have shown an opposite effect, in that winter holidays delay epidemic peaks[34–36], likely because children play a key role in household transmission, due to their reduced immune protection compared to adults, which is a notable difference between seasonal epidemics and a pandemic involving novel infectious agent, where initial population-wide immune protection is low. Besides vaccination, key NPIs recommended for influenza by the World Health Organization (WHO) include practicing good hygiene, such as handwashing, covering the mouth and nose when coughing or sneezing, and avoiding close contacts when feeling unwell[37]. Examining the effectiveness of NPIs implemented to prevent SARS-CoV-2 on influenza, the three measures with highest observed decline in importations were gathering limitations or mask-wearing

(applied together), travel bans or total border closures, and the closure of certain school levels[38]. This observation aligns with the understanding that influenza often spreads within schools, given lower immunity levels among children, in contrast to a pandemic, where a larger portion of the population lacks immunity.

While our study demonstrates the overall effectiveness of various NPIs in controlling the importation of SARS-CoV-2 into Germany, it is crucial to acknowledge the potential negative socioeconomic impacts associated with certain measures, such as the closure of workplaces and travel bans. Furthermore, since elimination of a pathogen like SARS-CoV-2 is not feasible in countries like Germany, factors beyond the importation of new lineages also play an important role for pandemic preparedness. In light of these considerations, our findings particularly highlight the efficacy of alternative interventions, specifically the implementation of freely available rapid testing (once available, since it needs to be developed for emerging pathogens) and the mandatory use of surgical or FFP2 masks in public places—two of the three most effective NPIs. These measures have proven to be both less harmful and highly effective in pandemic control, offering authorities a valuable and nuanced set of tools to navigate the challenging landscape of infectious disease management.

The substantial number of lineage importations into the German states with large populations and incidence numbers, as well as the prevalence of inter-state transmissions from these states highlights their relevance for maintaining the momentum of the pandemic in the third wave. These states most strongly contributed to the subsequent dissemination of imported SARS-CoV-2 lineages within the country, as most lineages were first observed in these states and transferred from thereon to others, and a reduction in this internal spread was evidently observable after application of internal NPIs. This aligns with a study of the UK[12], where cell phone-derived population mobility revealed in phylogeographic analysis the spread of SARS-CoV-2 lineages dominantly from the Greater London area.

Bayesian phylogenetic analyses of SARS-CoV-2 viral genomes have provided profound insights into pathogen evolution and dissemination during the pandemic[15,16,39]. We studied SARS-CoV-2 lineage importations into Germany using this approach with one of the most comprehensive sets of viral genomes analyzed to date, covering 1.8

million genome sequences. To mitigate temporal and regional, e.g., across countries and states, variations in the sequencing of infected patient samples, we used the three sampling strategies on these data, namely case ratio, 50:50, and 25:100 subsampling strategies. This allowed us to assess the potential effects of common sampling strategies on the results, such as e.g., the numbers of identified importation lineages, their sizes, and importation times across datasets. We identified many consistent properties across samplings, such as the very skewed abundance distribution of imported SARS-CoV-2 transmission lineages, with very few large lineages consistently observed, the number of imported lineages peaking after the German Christmas holidays, as well as the most populous German states having the most importations and contributing the most as a source of lineage spread within the country across states, and lineage importations being reduced following all twelve major NPIs that were implemented over the course of the third wave of the pandemic in Germany. Consistently, also the states with the fewest importations had the least relevance in spreading lineages within the country. Greater variation was observed for other properties, such as the TMRCAs and the absolute number of importation lineages. There was a tendency for an increase in the ratio of in-country to out-of-country samples and the absolute number of samples from within the country to lead to the merging of inferred importation lineages. This resulted in reduced absolute numbers of importation lineages and larger lineages with earlier TMRCAs. These findings suggest that caution is required when interpreting these properties.

By integrating phylogeographic and discrete trait analyzes to infer lineage importations with a longitudinal assessment of importation dynamics, we provide an analytical framework for exploring the correlations between environmental factors, such as holiday and non-pharmaceutical intervention at a population level to lineage dissemination. The effectiveness measure that we introduce allows us to systematically compare and quantify associations between NPIs and importations[7,11]. Through the design of this effectiveness measure, which assesses temporally local changes in the decline of lineage importations, potential longer-term effects on lineage importations, such as seasonality or altered transmissibility of individual lineages (B.1.1.7 versus B.1.177), have little effect. However, it is conceivable that confounding variables that coincide in time with a measure might contribute to the observed effects, such as behavioral changes resulting from increased concern or NPI-related fatigue in the population[40], and travel patterns and incidence rate in neighboring countries[11]. While certain NPIs may prove effective in specific situations, changes in circumstances, including shifts in the predominant lineage, can alter the apparent effectiveness. The B.1.1.7 lineage was substantially more transmissible than the previously circulating lineages[28], prompting the need to implement more restrictive NPIs. These initially dominated the case numbers during the studied time period and were subsequently supplanted by B.1.1.7 importation lineages. Finally, effects of NPIs may appear transient, as the number of importations also depends on case numbers outside of the country, which are not controlled by German NPI measures.

The SARS-CoV-2 pandemic was the first in which viral genomic surveillance was used to systematically monitor infection dynamics, enabling almost real-time insights into the evolution and dissemination of viral lineages worldwide, as well as allowing to link public health measures to these processes. Substantial efforts are being invested in improving preparedness in terms of rapid creation of vaccines, diagnostics, and therapeutics for future pandemics[41]. The results of this study and others demonstrate that establishing and analyzing the results of systematic viral genome surveillance is also essential for linking environmental factors to pathogen dissemination, informing about the relevance of non-pharmaceutical measures for the early days of an emerging pandemic.

## Methods

### Bayesian phylogenetic and phylogeographic analyses

All available SARS-CoV-2 genome sequences, 1,819,996 sequences overall, and the corresponding metadata were downloaded from the GISAID database on 2021-06-02[18,42]. Following established approaches, e.g., in the studies of Bbosa et al.[43], Bollen et al.[44], Hodcroft et al.[9], Lemey et al.[8], and Nemira et al.[7], we subsampled viral genome sequences using sampling dates, confirmed case numbers, and the country of origin, to account for varying data availability and ensure computational tractability in Bayesian phylogenetic and phylogeographic analyses. To assess the consistency of results across sampling schemes, we created two more datasets using different genome subsampling schemes for phylogeographic analyses[5,44]. While we sampled sequences proportional to the number of confirmed cases in each country for each week as our main subsampling strategy (called case ratio subsampling strategy); we sampled 100 sequences from Germany and 100 sequences from all other countries for each week (called 50:50 subsampling strategy); and finally, up to 25 sequences from each other country, if available, and 100 sequences from Germany for each week (called 25:100 subsampling strategy). From each of these datasets, we identified SARS-CoV-2 lineages imported into Germany, along with their estimated importation times. To infer a time-calibrated phylogenetic tree for each dataset, we performed a Bayesian analysis with Thorney BEAST version 0.1.1 (https://beast.community/thorney_beast). As input for the Thorney BEAST analysis we used a template tree obtained from collapsing short branches, an initial inferred maximum likelihood tree as well as the dates of the sample nodes. The Thorney BEAST analysis infers internal node heights as well as resolutions of polytomies and produces samples from the posterior distribution of the fully resolved phylogenies. Importations of SARS-CoV-2 lineages into Germany were identified as described by du Plessis et al.[5], corresponding to lineages that include at least one German viral genome and descend from an ancestral lineage inferred to have circulated outside of Germany (Fig. 6A). After performing a Bayesian DTA step, which assigns either a Germany or non-Germany label to the internal nodes of the tree, and calculating the Maximum Clade Credibility (MCC) tree, the importation lineages were identified as maximal connected subsets of nodes labeled as Germany. Following the inference of importation lineages, we examined the lineage diversity within states using two metrics: the Shannon Index (SI) and lineage evenness[22,23]. To infer the spread of SARS-CoV-2 importation lineages within Germany between the 16 German states, we applied a 16-state Bayesian Discrete Trait Analysis (DTA) method, each DTA state corresponding to one state of Germany, to the phylogenies of the inferred importation lineages using BEAST (Fig. 6B). Further details on all steps are described in the Supplementary Methods.

When referring to an importation lineage, the term lineage is used for brevity throughout, while Pangolin lineage assignments are always explicitly referred to as such without explicitly mentioning importation for brevity.

### Relating lineage importation dynamics to nonpharmaceutical interventions

To assess the effect of each nonpharmaceutical intervention on lineage importations, we defined the smoothed importation frequency (SIF) as follows (more details in the Supplementary Methods, Definition of the effectiveness measures): Let $s$ be a smoothing parameter (to be used later), and set it to 5. Then, the SIF for each day is defined as a weighted sum of the number of importation events that happened on the current day and the next $2s - 2$ days. The weight coefficients $w(j)$ are defined as $w(j) = j$ for $1 <= j <= s$ and $w(j) = 2s - j$ for $s + 1 <= j <= 2s - 1$, creating a triangular graph with a peak at $w(s)$. The measure is calculated as the sum of the number of importation

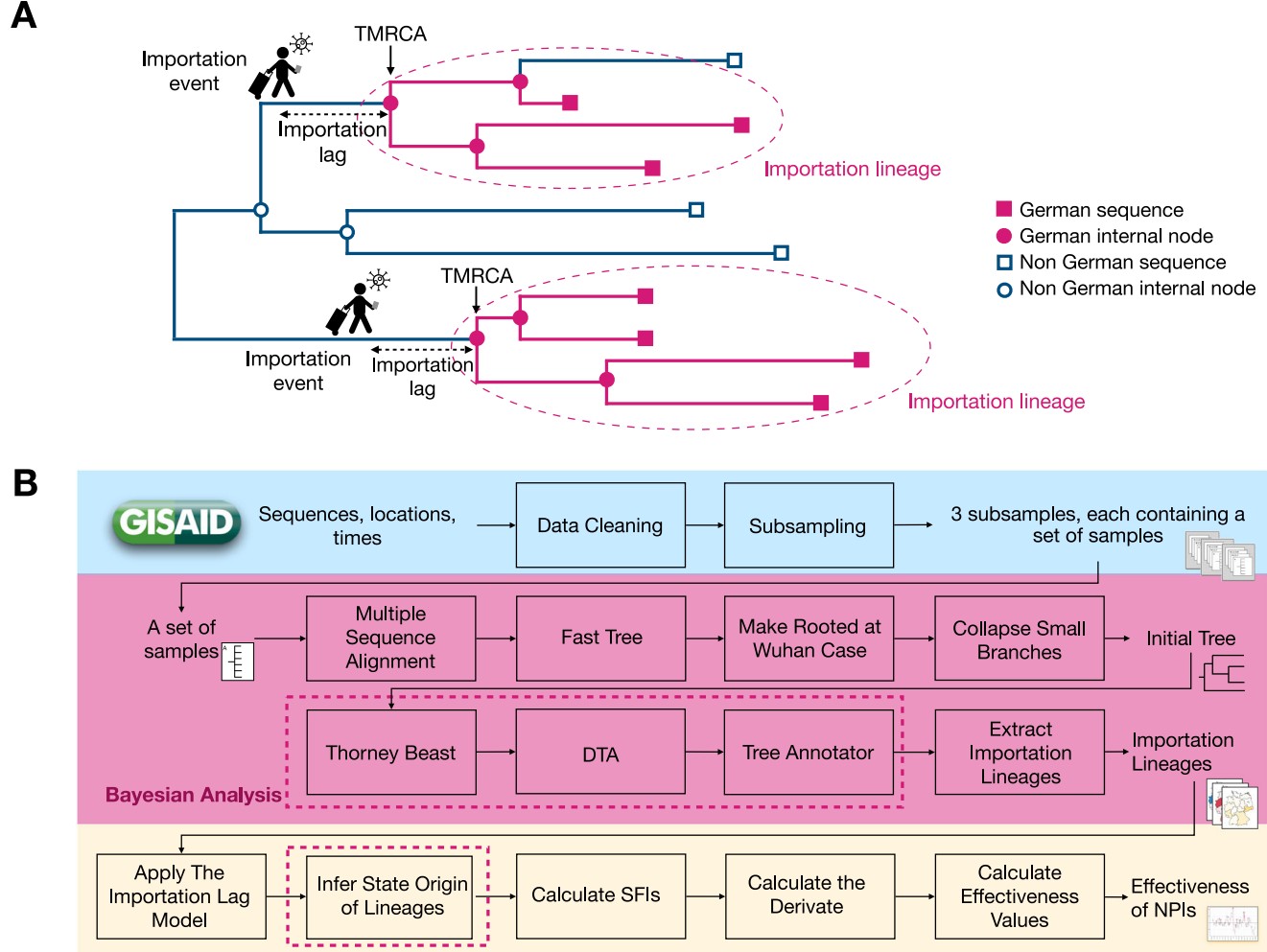

**Fig. 6 | Schematic overview of the method. A** Definition of importation lineages on an inferred phylogeny (orange indicates ancestral lineages and sampled isolates circulating in Germany, blue those found outside of Germany). An importation lineage is defined as a subtree with all nodes (lineages) from Germany that descend directly from an ancestral lineage (node) inferred to have circulated outside of Germany. The time of the most recent common ancestor (TMRCA) is the inferred time of the first junction in an importation lineage. The importation lag is the difference in time between the entrance of an infected person, i.e., the importation event, to Germany and its TMRCA. **B** Analytic workflow overview, with (i) genome data selection via sampling (blue), (ii) phylogenetic and phylogeographic analyses of importation lineages using a Bayesian framework (pink), and (iii) assessment of importation lineage dynamics together with NPIs implemented in Germany during the third wave (yellow).

events on the *j*-th day multiplied by the weight $w(j)$, which produces a smooth measure with a peak that focuses around the *s*-th following day.

We gathered information about 119 NPIs implemented in Germany from published sources (Table S4)[24–26,45], summarized the national and local NPIs into 12 major NPIs, and classified them as internal NPIs, border control NPIs, and NPIs including both internal and border control measures. Internal measures include workplace closing, restrictions on gathering, restrictions on internal movements, canceling public events, schools closing, obligation on surgical/FFP2 mask-wearing, and availability of free antigen tests. We studied the effectiveness of NPIs on lineage importations and spread, first for the country and then for the states with the highest importation activities and population sizes.

### Declaration of generative AI and AI-assisted technologies in the writing process

The authors used ChatGPT to improve the language of the text and to shorten the abstract. After using this tool/service, the authors reviewed and edited the content as needed and take full responsibility for the content of the publication.

### Reporting summary

Further information on research design is available in the Nature Portfolio Reporting Summary linked to this article.

## Data availability

SARS-CoV-2 isolate genome sequence data and metadata were downloaded from GISAID and are accessible through their website. The GISAID Accession IDs for SARS-CoV-2 genome sequences are available at https://doi.org/10.5281/zenodo.10963104. All the analyzed data generated in this study are available at https://github.com/hzi-bifo/covid-germany-mcmc. The regulations on non-pharmaceutical interventions were compiled from multiple sources: the corona_tscs dataset: https://doi.org/10.5281/zenodo.5201766, the Oxford COVID-19 Government Response Tracker https://doi.org/10.1038/s41562-021-01079-8, and information provided by Germany's federal government https://www.deutschland.de/en/news/

coronavirus-in-germany-information and https://www.deutschland.de/en/news/german-federal-government-informs-about-the-corona-crisis.

## Code availability

All the codes are publicly available at https://github.com/hzi-bifo/covid-germany-mcmc.

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

## Acknowledgements

We gratefully acknowledge all data contributors, i.e., the authors and their originating laboratories responsible for obtaining the specimens, and their submitting laboratories for generating the genetic sequence and metadata and sharing via the GISAID Initiative, on which this research is based. The authors gratefully acknowledge funding by the German Center for Infection Research (DZIF project number TI 12.002_00) and funding by the German Research Foundation (DFG), for the Excellence Cluster RESIST EXC 2155 project number 390874280, as well as all data contributors, i.e, the authors and their originating laboratories responsible for obtaining the specimens, and their sub-mitting laboratories for generating the genetic sequence and metadata and sharing via the GISAID Initiative, on which this research is based. A.W., S.Ö. and D.K. were supported by the Max Planck Society.

## Author contributions

S.G., M.H.F.A. developed the codes. A.M.H. conceived the study. S.G., M.H.F.A., A.R., A.W., S.O., D.K., and A.M.H. analyzed the data. S.G., M.H.F.A., A.R., and A.M.H. wrote the article with comments by all authors.

## Funding

## Competing interests

The authors declare no competing interests.
