## [Peer Review File · Nature Communications]

Importations of SARS-CoV-2 lineages decline after nonpharmaceutical interventions in phylogeographic analysesReviewers' Comments:

Reviewer #1:

Remarks to the Author:

In their manuscript Goliaei and colleagues use large-scale phylodynamics to assess the efficacy of nonpharmaceutical interventions against SARS-CoV-2 in Germany. The authors find that multiple internal and travel based restrictions reduced the rate of importation into Germany and the movement of lineages between German states. The authors also find that the majority of importations arrived in three, populous states and that these states acted as sources of further spread around the country.

This is a nice manuscript that describes the transmission-lineage dynamics early in the pandemic. The authors carefully explore the impact of sampling bias and ensure the robustness of their results through multiple subsampling schemes. I have only one general concern with the results that I believe can be easily addressed. Throughout the manuscript the text communicates a causal relationship between NPIs and decreased lineage importations. This relationship could be strengthened.

Concerns:

The smoothed importation frequency for a given day is defined as a function of the number of importations over a nine day period. It is not clear from the text if this is the nine days preceding or following the day of interest, but it does not seem to be centered on the day of interest. The authors should provide a justification for this definition, and how it affects their comparisons of SIF throughout the study period? Is the efficacy of NPIs affected by this definition.

How sensitive is the DEM to outliers? It seems this metric compares extreme observations between two time periods. (Maybe this accounted for by the smoothing function?)

Possibly related to the points above, the importation lag (Figure 1) will affect the observed timing of importations. What is the estimated importation lag throughout the study period? A importation lag model is referenced in the supplementary materials but I didn't see the details (sorry if I overlooked them).

The effectiveness of a given NPI is determined by the SIF immediately following an intervention's implementation. However, the number of importations is dynamic throughout the study period often rebounding immediately after an intervention and in many cases falling in weeks where no new interventions were implemented. If the NPI successfully limits lineage import and spread why is the effect transient? Do weeks following the implementation of NPIs have a greater negative change in SIF than expected given the noisiness of the importation dynamics?

The largest decrease in importation intensity occurs after FFP2 masks were made mandatory. But this also coincides with the period after Christmas which saw the largest import rate, likely due to holiday travel. It's not clear to what extent the decrease can be attributed to mask wearing as opposed to less travel after a period of high-travel.

Figure 5A and 6A show a steady decline in importations and inter-state movements near the end of the study period. Can these trends be attributed to the effect of the NPIs or are they driven mostly by the dataset being right truncated?

The results on internal movements in figure 6 is very cool. Are there similar increases in importations following the relaxation of NPIs in February? (a lack of an increase might be caused by limited cases if they were falling during this time period.)

The role of importations and internal migrations could be better placed in context of wave 3. For instance, how do these events and their timings correspond to cases counts throughout the study period? It could also be interesting to see how the rate of detecting new lineages (and the rate of lineage extinction) changes overtime and in response to NPI as well as the arrival of Alpha.

Very Minor Concerns:

Line 235: The date format here is ambiguous. It would be more clear to use the YYYY-MM-DD format as in Figure 2.

Reviewer #2:

Remarks to the Author:

This manuscript is a nice application of phylogeography to a very relevant public health issue. I really appreciate this work as it can be used widely in guiding future public health responses, especially promoting the use of free rapid antigen tests and mask-wearing for preventing SARS-CoV-2 and other respiratory viruses. Overall, I think this is a good manuscript and needs very little changes. Great work, and I look forward to your final manuscript!

Major comments

Methods

- why 9 days for SIF? Why day 5 for the peak?
- Figure 2A. Is this for the total time period?
- line 169/170: I apologise but I do not understand what you mean by splitting lineages, or what you are referring to when you say truly circulating lineages. Could you please define further?
- was there any difference between states in Germany for sequencing proportions?
- When you are discussing the airports in Line 196 paragraph, I would also suggest making a clear point that with airport and air travel hubs and respiratory viruses, unless the airport is in a densely populated city, they are unlikely to be the area where the viruses is seeded due to the infection delay. Although many people may be infected during air travel, they are more likely to infect others at their final destination. To pull those dynamics apart though, you would need travel history metadata.
- Could you please include a table describing the NPIs you refer to in text! I think it is a really important thing to be able to see as you are reading the results.
- I understand you are communicating many things in Figure 4, but I found it too confusing and took too long to interpret. I don't have any easy-fix recommendations for how to make it a better figure (apologies!), but at present, I don't think it is communicating what you need to the reader. Perhaps if you added a legend for the colours, and condensed some of the symbols? I also think the NPIs need to be written out or at least have some kind of description here and in Figure 5B.

Discussion

- When you discuss lockdowns, could you please define exactly what lockdown means in this context, as readers from different countries will have their own interpretations of what a lockdown is (in terms of intensity, NPIs, period).
- When you discuss flu (line 308), could you maybe discuss further (i.e, what control measures are used to prevent flu, how could COVID be different?)
- you could probably discuss the lack of association of importations with airports in line 313/314
- I would like to see you really highlight how effective non-invasive NPIs appear to be in controlling/preventing SARS-CoV-2! The travel bans that have been implemented in various countries have had harmful effects, both socially, economically and culturally. Yet, unfortunately, it will probably keep happening - but studies like this really show how we can reduce infectious disease better with

simple measures.

Minor comments

Abstract:

- I think that because mask-wearing was such an important role in effective NPI use, you should highlight that in your second sentence, perhaps in between "antigenic testing, or travel restrictions". I.e.: "antigenic testing, physical transmission barriers (e.g. medical/surgical masks) or travel restrictions".
- there are two commas that are not needed, in line 25 and 36, respectively between third, pandemic and few, negative.

Introduction

- a few minor grammatical changes I would recommend include line 57, changing it to: pandemic [4]. For example, previous studies suggest that it was transported from Hubei...
- Define UK as United Kingdom at first use, then change all U.K. to UK
- Line 85: I think that using "first emerged" is slightly controversial in terms of SARS-CoV-2, and I prefer using "first detected" as it is more accurate.
- You have defined NPIs but still define it a few times in the manuscript, where you could use the acronym, eg. line 91.

Methods

- Could you please define the dates for the Christmas holidays in text and in Figure 2.
- Just a suggestion - I would use constantly or invariably instead of "Stably"

Discussion

- Line 342 is a massive sentence - could be split into 2-3 smaller sentences.

Reviewer #3:

Remarks to the Author:

This study attempts to quantify the effectiveness of different non-pharmaceutical interventions (NPIs) on the rate and distribution of SARS-CoV-2 lineage importation events. This is an enormously difficult problem that is limited in part by the challenge of adjusting for variation in sampling rates over time and among regions not only within Germany (the focal country), but also sampling in other countries from which importation events must be inferred. The authors employ a recently developed Bayesian method (Thorney BEAST) to generate a random sample of trees from the posterior distribution. This is an important feature of the analysis, but the main text of the manuscript lacks any explanation of how this method works, and there is limited mention in the Supplement.

From my quick reading of the associated literature, this method appears to constrain the tree topology to an input tree, instead resampling internal node heights based on the tip dates and a clock model. This implies that importations are fixed features of the input template tree. I'm concerned that reconstructing importations have a substantial level of uncertainty. SARS-CoV-2 phylogenies are difficult to resolve, resulting in many polytomies and nodes with low support. I think it would be important not only to quantify this uncertainty, but also to propagate it forward to measuring the impacts of different NPIs. Instead, key quantities such as daily effective measures (DEMs) are being reported without any error (e.g., line 244). This also applies to reconstructing the spread of lineages between countries and states. If the authors are not going to propagate uncertainty at all, then what is the point of using Bayesian sampling in the first place? It would have been much faster to rescale the tree in time by maximum likelihood, wouldn't it?

Determining the effect of different NPIs on the variation in the rates of NPI among regions and over

time is an exceedingly difficult problem, for instance because of confounding between the effects of different interventions. Each NPI category corresponds to a time period in which multiple NPIs of varying mechanisms and distribution are in effect. Moreover, there are a limited number of changes in interventions over time which limit our ability to evaluate causal relationships between each intervention and the NPI rate as a geographically- and temporally-structured outcome. Nevertheless, I found the authors' approach to be feasible and a reasonable advance on similar work in the literature.

Specific comments

* p.4, line 84: "The third infection wave in Europe during spring 2021 [...]" The authors also talk about third and fourth waves in the preceding paragraph, which is confusing. Are they making a distinction between waves in east Asia and Europe? If so, it would be helpful for this to be spelt out clearly.

* p4, lines 103-104: "Following established approaches [...], we subsampled viral genome sequences [...]" There are actually more methodological details provided in the Results (lines 158-161) and Discussion (lines 326-334) sections of the manuscript than in this Methods section, which is not helpful. Why can't this material be collected into Methods?

* line 115, Shannon Index and lineage evenness are not described in the Supplement.

* line 116, why 16 states? what are these character states being mapped to?

* line 117, please define "DTA" at first use.

* line 120 and onward, it would be helpful to provide some rationale for this smoothing function, i.e., why does it peak at four days post reference time point? This sentence: "We determined the effectiveness of an NPI considering [...]" is difficult to parse and would be easier to understand with a mathematical formula. I'm having trouble arriving at a 22 day time interval. Given NPI at time 0, the previous seven days takes our time line out to -7 and the next seven days to +7. At time +7, the SIF is calculated from time points +7 to +15.

* Figure 1A does not render well in greyscale - this would be resolved in part by using open and closed circles/squares in addition to colour.

* line 150, "lineage[s]"

* I find the use of "lineage" to refer to both importations and PANGO designations to be somewhat confusing, e.g., repeated use in first line of Figure 2 legend; also line 207, "total number of lineages" refers to what?

* line 193, what do you mean by "CC", correlation coefficient? Please define at first use.

* line 202-204, "We hypothesize that the state population is more reflective of the ultimate destination of these travels, while airports serve as intermediate stops." This seems like a bit of a trivial statement - most people immediately leave the airport upon arrival. I surmise that the observed trend is due to the amplification of newly arrived lineages in large susceptible populations.

* line 227-228, "we gathered information about 4,000 national and local NPIs and summarized them into 12 major NPIs by date" - but earlier you said that you examined 110 NPIs (line 133)!

* line 237, I find it difficult to keep track of the different major NPIs with the N1-N12 numbering system, but I can't think of a better alternative.

* line 348, "By integrating a phylogenetic and phylogeographic analysis [...]" This amalgamation seems awkward to me, because phylogeography is essentially a specialized application of phylogenetics. I guess the authors are trying to allude to a hierarchical aspect of their analysis, i.e., ancestral state reconstruction model applied to phylogenies?

* line 355-359, "However, it is conceivable that confounding variables that coincide in time with a measure might contribute to the observed effects [...]" There is almost surely some confounding given the complexity of this system. What would be the expected effect of some of the most likely and strongest confounders?

* Supplement lines 12-13, "Low-quality samples, corresponding to sequences shorter than 28,000 bp or with more than 1000 ambiguous bases, were removed as well [...]" What was the distribution of the number of ambiguous bases? Please provide a histogram summarizing this distribution. Sequences with substantial numbers of ambiguous base calls may be problematic for resolving phylogenies.

* Supplement lines 16-18, "Bayesian phylogeography is a technique to study [...] pathogen spread, allowing integration of genomic information with different types of metadata and epidemiological spread models." I think this definition is placing a strong emphasis on using a demographic/epidemiological model as a tree prior, but I don't think that Bayesian phylogeography is not restricted to this approach.

* Supplement lines 24-28, this subsampling and stratification of data to a (relatively) small number of PANGO lineages (12) is a really important aspect of the study. What proportion of samples was discarded by restricting the analysis to this set of PANGO lineages? Are "derived" lineages included in these samples, or are these "pure" lineage sets? For example, are you including any of Q.1 (B.1.1.7.1), or B.1.36.*? In addition, what PANGO lineage assignments are you using? Are you using the classifications provided by the GISAID database, or re-classifying sequences de novo? Which classifier and version of lineage definitions are you using?

* Supplement line 29, please clarify what you mean by "invalid dates".

* Supplement lines 53-58, you've named the subsampling strategies here quite clearly, but it would be helpful to provide at least a brief explanation in the main text.

* Supplement lines 50-51, "These further datasets contain 2689/2691 sequences from Germany and 1932/25,107 non-German sequences, respectively." It's not clear how you are using the forward slash here. We have to assume that the second number in each pair represents the third sampling strategy (25:100).

* Supplement lines 75-76, there are quite a few BEAST settings missing here. Are you using exactly the same substitution model and prior hyperparameters used in du Plessis et al.?

* Supplement line 79, "Convergence was ensured using LogAnalyser [...]" Convergence of a chain sample to the posterior distribution is never ensured.

* Supplement line 97, this would be a good spot to provide formulae for Shannon Index and lineage evenness.

* Supplement line 104, what are the 16 states?

* Supplement line 85, can you provide a reference for the maximum clade credibility tree? The majority of readers will not be familiar with this concept.

Reviewer #1

In their manuscript Goliaei and colleagues use large-scale phylodynamics to assess the efficacy of nonpharmaceutical interventions against SARS-CoV-2 in Germany. The authors find that multiple internal and travel based restrictions reduced the rate of importation into Germany and the movement of lineages between German states. The authors also find that the majority of importations arrived in three, populous states and that these states acted as sources of further spread around the country.

This is a nice manuscript that describes the transmission-lineage dynamics early in the pandemic. The authors carefully explore the impact of sampling bias and ensure the robustness of their results through multiple subsampling schemes. I have only one general concern with the results that I believe can be easily addressed. Throughout the manuscript the text communicates a causal relationship between NPIs and decreased lineage importations. This relationship could be strengthened.

We thank reviewer 1 for the positive feedback and providing highly constructive and insightful comments for our work. These helped us to further strengthen the relationship between NPIs and decreased lineage importations. It is also important to note that throughout the manuscript the use of causal wording such as “NPIs reducing lineage importations” is avoided, due to the challenges of their exact quantification, and instead describes the temporal order of events, such as that lineage importations declined following NPIs. Next, we outline how we have followed the reviewers suggestion to provide further support for the possibility of causal relationships.

Concerns:

1. *The smoothed importation frequency for a given day is defined as a function of the number of importations over a nine day period. It is not clear from the text if this is the nine days preceding or following the day of interest, but it does not seem to be centered on the day of interest. The authors should provide a justification for this definition, and how it affects their comparisons of SIF throughout the study period? Is the efficacy of NPIs affected by this definition.*

We thank the reviewer for this comment. In response, we clarified the definition of the smoothed importation frequency in the Methods section to (Lines 141-149):

“To assess the effect of each nonpharmaceutical intervention on lineage importations, we defined the smoothed importation frequency (SIF) as follows (more details in the Supplementary Methods, Definition of the effectiveness measures): Let s be a smoothing parameter (to be used later), and set it to 5. Then, the SIF for each day is defined as a weighted sum of the number of importation events that happened on the current day and the next $2s-2$ days. The weight coefficients $w(j)$ are defined as $w(j) = j$ for $1 \leq j \leq s$ and $w(j) = 2s-j$ for $s+1 \leq j \leq 2s-1$, creating a triangular graph with a peak at $w(s)$. The measure is calculated as the sum of the number of importation events on the j -th day multiplied by the weight $w(j)$, which produces a smooth measure with a peak that focuses around the s -th following day.”

Second, we added the formula for the smoothed importation frequency and daily effectiveness measure to the supplementary materials (Lines 133-143):

“Definition of the effectiveness measures

Let s be the smoothing parameter and $\mu(i)$ be the importation frequency for day i . Then the measures SIF $\sigma(i)$, change rate of SIF $\sigma'(i)$, and effectiveness $\eta(i)$ on day i are calculated as follows:

$$\sigma(i) = \sum_{j=1}^s j \mu(i+j-1) + \sum_{j=s+1}^{2s-1} (2s-j) \mu(i+j-1)$$

$$\sigma'(i) = \sigma(i) - \sigma(i-1)$$

$$\eta(i) = \max_{t \in \{-6, \dots, 6\}} \left(\sigma'(i+t) - \min_{k \in \{\max(0,t), \dots, 6\}} \sigma'(i+k) \right)$$

Based on these formulas, $\eta(i)$ depends on $\sigma'(i-6), \dots, \sigma'(i+6)$, which depend on $\sigma(i-7), \dots, \sigma(i+6)$, which depend on the number of importations in $2s+12$ consecutive days: $\mu(i-7), \dots, \mu(i+2s+4)$. Here, the smoothing parameter is $s=5$, and the effectiveness depends on the number of importations in 22 consecutive days.”

Third, to further clarify the reasoning for the definition of the smoothed importation frequency and selection of value 5 for the smoothing parameter s , we added this paragraph to the supplementary materials (Lines 144-154):

“We aimed to limit the effectiveness assessment to a local scope of at most three weeks, to eliminate seasonal effects (as shown in (Gross et al. 2020) in 20 days a hard lockdown stabilizes the number of confirmed cases). By selecting a smoothing parameter (the parameter “ s ” in the definition of the effectiveness measures) not greater than five days, importations from more than three weeks later do not affect the calculated effectiveness of an NPI. Furthermore, to distinguish the effect of different NPIs, we maintained the smoothing parameter to be less than the minimum distance of six days between two consecutive NPIs. Therefore, we opted for the maximum possible value of five for the smoothing parameter to obtain a well-smoothed curve. In addition, when an NPI is implemented, it takes a few days for its effects to be observed between different states and cities. Therefore, we designed the triangular weighting function, to increment slightly every day from the actual day, and then decrease.”

Finally, to investigate the sensitivity of the SIF and the final results to the smoothing parameter, we performed further experiments and added these results to the supplementary materials (Lines 166-173):

“Sensitivity analysis of the smoothed importation frequency and effectiveness measures

To investigate the sensitivity of the final results to the smoothing parameter, we assessed the SIF measure for various values for the smoothing parameter. Increasing the smoothing parameter results in a smoother curve, while decreasing it reveals sharper changes in the curve. The actual SIF values also increase with higher smoothing parameters, for which a normalized chart is also provided. Notable peaks associated with effective NPIs, such as N8[mask] and N11[free test], are still discernible in all the curves.”

Fig. S12 - SIF for different values of the smoothing parameter. A) SIF curves calculated for different smoothing parameters. **B)** SIF curves normalized by the smoothing parameter squared. While increasing the smoothing parameter leads to higher absolute values of the SIFs, the normalized values maintain a relatively consistent average across different smoothing parameters. In this study, a smoothing parameter of 5 is employed.

We added the following sentence to the main text (Lines 307-310):

“Analysis of the sensitivity of the SIF and effectiveness measures show the robustness of the results with respect to minor changes in the smoothing parameter (Supplementary Materials, Sensitivity analysis of the smoothed importation frequency and effectiveness measures, **Fig. S12**).”

2. How sensitive is the DEM to outliers? It seems this metric compares extreme observations between two time periods. (Maybe this accounted for by the smoothing function?)

*We thank the reviewer for this comment. To clarify the sensitivity of the DEM (daily effectiveness measure, shown in **Fig. 5B** in the main text) to outliers, we added the following paragraph to the supplementary materials (section **Sensitivity analysis of the smoothed importation frequency and effectiveness measures**) (Lines 174-179):*

“Two steps of the method contribute in particular to reducing the sensitivity of the results to outliers present in the primary input dataset. 1) Since the final effectiveness values are derived based on samples from a posterior distribution of a Bayesian inference process, the results are based on multiple samples, diminishing the variance of inferred parameters compared to a naive method that relies on single inferred variables. 2) Additionally, we applied a smoothing function to the lineage importation curve to further mitigate the impact of outliers.”

3. Possibly related to the points above, the importation lag (Figure 1) will affect the observed timing of importations. What is the estimated importation lag throughout the study period? A importation lag model is referenced in the supplementary materials but I didn't see the details (sorry if I overlooked them).

*We thank the reviewer for this comment, which led us to clarify the stability of daily effectiveness measures under changes in importation lag. We added the following paragraph (section **Sensitivity analysis of the smoothed importation frequency and effectiveness measures**), table, and figure to the supplementary materials (Lines 180-200):*

“We incorporated a fixed lag time of five days into our analysis. The concept of the importation lag, utilized for determining the actual importation of lineages, was introduced by du Plessis et al. (du Plessis et al. 2021). The lag model they propose exhibits an inverse dependency on the lineage size and the parameters were determined by fitting the importation model to lineages representing a smaller number of confirmed cases, compared to the lineages of the current study. We set the duration of the lag as five days as determined by Nadeau et al. (Nadeau et al. 2023), since the lineage size is the determining parameter for the lag-model proposed by (du Plessis et al. 2021), and the lineage size inferred in our work surpasses that of du Plessis and aligns more closely with the lineage sizes in Nadeau et al. Notably, the final results, especially for the two most effective NPIs N11[free test] and N8[mask], are stable under slight changes (from 1-6 days) of the importation lag (**Table S5**), which reflects the range of importation lags for lineages of size 5 or larger used in both studies. For higher importation lags ranging from 7 to 10 days, N8[mask] and N9[variant travelers ban] exhibit the highest effectiveness, while for a lag of 7 days, N11[free test] ranks as the third most effective, with a DEM value only one point less than those of N8[mask] and N9[variant travelers ban]. It's important to note that N9[variant travelers ban] was implemented only 6 days after N8[mask], thus some confounding effects cannot be ruled out. As for every one day change in the importation lag (decrease or increase), the SIF curve is shifted one day (left or right, **Fig. S13**), effects of preceding measures will partly be attributed to following ones, if lag values

are set too high. The low effectiveness of variant traveler ban N10[border closure] indicates a lower relevance of traveler bans, such as N9[variant travelers ban].”

Table S5 - Effectiveness for NPIs for different lag models. Each column represents the effectiveness of an NPI with respect to a specific importation lag value, ranging from 0 to 10 days. In each column, the three NPIs with the highest effectiveness are highlighted in bold. A lag time of 5 is the main model used in this study.

Date	NPI	Effectiveness value for importation lag = 0 to 10 days										
		0	1	2	3	4	5 (this study)	6	7	8	9	10
2020-10-19	N1	8	8	7	14	17	17	17	17	17	17	15
2020-11-02	N2	6	6	5	7	10	11	14	14	14	14	11
2020-11-08	N3	8	8	6	6	6	6	6	6	5	7	10
2020-11-30	N4	9	14	14	14	14	14	14	14	14	12	12
2020-12-16	N5	16	16	16	16	16	16	16	16	11	6	6
2020-12-22	N6	13	13	13	13	8	10	16	16	16	16	16
2021-01-11	N7	11	11	16	16	16	16	16	16	16	16	15
2021-01-24	N8	20	24	24	24	24	24	24	24	24	18	18
2021-01-30	N9	10	10	10	10	10	10	20	24	24	24	24
2021-02-14	N10	23	23	23	17	12	12	12	12	12	12	12
2021-03-13	N11	31	31	31	31	31	31	31	31	23	17	15
2021-03-30	N12	12	12	11	9	12	12	12	12	12	12	12

Fig. S13 - The smoothed importation frequency (SIF) through time for different importation lag values. SIF through time, for different values of importation lag (0, 5, and 10 days).

We added the following sentence to the main text (Lines 310-311):

“Furthermore, N11[free test] and N8[mask] are consistently two most effective NPIs for changes in importation lag, from importation lag be set from 1 to 6 days (**Fig. S13, Table S5**).”

4. *The effectiveness of a given NPI is determined by the SIF immediately following a intervention's implementation. However, the number of importations is dynamic throughout the study period often rebounding immediately after an intervention and in many cases falling in weeks where no new interventions were implemented. If the NPI successfully limits lineage import and spread why is the effect transient? Do weeks following the implementation of NPIs have a greater negative change in SIF than expected given the noisiness of the importation dynamics?*

We agree with the reviewer that, in many cases, the number of importations increases within a few weeks, or even a week, after the application of an NPI, or decreases when no new NPI is implemented. The number of importations is influenced by various factors in the short or long term, including seasonal effects, changes in the abundance of different lineages, and the number of COVID-19 cases in other countries. These factors may contribute to fluctuations in the number of importations even when no major NPI is implemented.

For short-term increases observed following the application of an NPI, it's possible that the implementation of an NPI coincides with a period when the number of importations is already increasing due to other factors. This is particularly relevant when NPIs are suggested based on the current adverse situation of the pandemic inside the country. In such cases, the implementation of an NPI may help control the ongoing increase in importations, but the increasing factor might still be present, and its effect is still observable but seems to be delayed.

This is one of the strengths of the SIF, that it allows to distinguish behavior immediately following an NPI from changes caused by long-term effects such as that of seasonality. In response to this comment, we extended the paragraph about the definition of the effectiveness measure to elaborate (Lines 440-458):

“By integrating phylogeographic and discrete trait analyzes to infer lineage importations with a longitudinal assessment of importation dynamics, we provide an analytical framework for exploring the correlations between environmental factors, such as holiday and nonpharmaceutical intervention at a population level to lineage dissemination. The effectiveness measure that we introduce allows us to systematically compare and quantify associations between NPIs and importations [7,11]. Through the design of this effectiveness measure, which assesses temporally local changes in the decline of lineage importations, potential longer-term effects on lineage importations, such as seasonality or altered transmissibility of individual lineages (B.1.1.7 versus B.1.177), have little effect. However, it is conceivable that confounding variables that coincide in time with a measure might contribute to the observed effects, such as behavioral changes resulting from increased concern or NPI-related fatigue in the population [44], and travel patterns and incidence rate in neighboring countries [11]. While certain NPIs may prove effective in specific situations, changes in circumstances, including shifts in the predominant lineage, can alter the apparent effectiveness. The B.1.1.7 lineage was substantially more

transmissible than the previously circulating lineages [32], prompting the need to implement more restrictive NPIs. These initially dominated the case numbers during the studied time period and were subsequently supplanted by B.1.1.7 importation lineages. Finally, effects of NPIs may appear transient, as the number of importations also depends on case numbers outside of the country, which are not controlled by German NPI measures.”

5. *The largest decrease in importation intensity occurs after FFP2 masks were made mandatory. But this also coincides with the period after Christmas which saw the largest import rate, likely due to holiday travel. It's not clear to what extent the decrease can be attributed to mask wearing as opposed to less travel after a period of high-travel.*

*We appreciate the reviewer's thorough evaluation and insightful observation. It's important to note that fully disentangling the impact of one single factor from several complex influencing factors on importations is a challenging problem. However, certain aspects enhance our confidence in attributing the observed reduction in importations mainly to the implementation of NPI N8[mask] on 2021-01-24, rather than solely to the conclusion of the Christmas holidays. To illustrate this challenge and highlight the aspects demonstrating a potential connection between NPI N8[mask] and the subsequent decrease in the number of importations we add the following paragraph to the main text, section **Lineage importations are reduced after nonpharmaceutical interventions** (Lines 276-291), and the following figure to the supplementary materials.*

“While disentangling the separate factors affecting importations is a challenging task, several factors enhance confidence in attributing the observed reduction in importations mainly to the implementation of NPI N8[mask] on January 24 rather than solely to the conclusion of the Christmas holidays. To understand the extent to which the ending Christmas holidays' have an effect on importations, we examined the importation pattern in the three most populous states. We exclude Bavaria from this observation, since Bavaria first adapted the internal 15km movement ban on 2021-01-11, then relaxed it a few days later. Examining lineage importations in Baden-Württemberg, a highly populated state that did not implement the internal movement ban in NPI N7[15km movement ban], allowed us to analyze the effect of ending Christmas holiday on importations with less NPI influence. The change rate (derivative) of SIF began to increase on 2021-01-09, marking the end of the Christmas school holidays, and decreased on 2021-01-16, occurring nine days before the implementation of N8[mask] on 2021-01-25 (**Fig. S10**). The same pattern is observable for North-Rhine Westphalia, where the change rate of SIF began to increase on 2021-01-09, and decreased on 2021-01-15, which is ten days before the implementation of N8[mask] in North-Rhine Westphalia on 2021-01-25. Consequently, we deduce that holiday travels had already decreased before implementation of N8[mask].”

*We also added the following text to Supplementary material, section **supplementary results** (Lines 201-213):*

“Analysis of derivative of SIF in the three states with highest population in Germany

In Baden-Württemberg (BW), where the Christmas holidays concluded on January 9, 2021, the derivative of SIF began to increase on 2021-01-10, reaching 0 on 2021-01-12, indicating that SIF changed to increasing (**Fig. S10**). The derivative of SIF peaked at 5 on 2021-01-16, then started decreasing, indicating a reduction in the change rate of importations in Baden-Württemberg. The derivative returned to 0 on 2021-01-09, indicating a decrease in SIF, 6 days before the implementation of N8[mask] on 2021-01-25. In North Rhine-Westphalia (NW), the derivative of SIF began to increase on 2021-01-10, reaching 0 on 2021-01-11, marking the initiation of SIF increase at the end of the Christmas holidays. On 2021-01-16, the derivative peaked and started decreasing.

Subsequently, on 2021-01-17, the derivative again reached 0, signifying the commencement of SIF decrease in North Rhine-Westphalia, 7 days before the implementation of N8[*mask*] in this state.”

Fig. S10 - Derivative of SIF for January 2021. BY, BW, and NW in the label of the arrows stands for Bavaria, Baden-Württemberg, and North Rhine-Westphalia, respectively. This figure illustrates the derivative of the smoothed importation frequency for the three most populated states in Germany. A negative derivative value indicates a decrease in SIF, 0 implies that SIF has reached a plateau, and a positive derivative value indicates an increase in SIF.

6. Figure 5A and 6A show a steady decline in importations and inter-state movements near the end of the study period. Can these trends be attributed to the effect of the NPIs or are they driven mostly by the dataset being right truncated?

To clarify, we added the following paragraph to the **Result** section (Lines 351-353):

“Notably, the steady decline of importations and inter-state movements evident at the very end of the study period is linked to the end of our study period, as the dataset we used includes data available until June 2nd, 2021 in GISAID (**Fig. S20**).”

Fig. S20 - The number of sequences in the dataset of the current study versus the sequences in the GISAID dataset as of 2021-08-30. A) The ratio of the seven-day moving average number of sequences in the GISAID dataset as of 2021-06-02 (the dataset upon which all inferences in this study are based) to the number of sequences in the GISAID dataset as of 2021-08-30. The chart includes all sequences with a valid collection date format (depicted by the red line) as well as the same ratio for the sequences from Germany with a valid collection date format (illustrated by the cyan line). **B)** The actual number of sequences in the GISAID dataset as of 2021-06-02 and 2021-08-30, considering the same conditions as outlined in subfigure A. Since it takes some time for the data to be incorporated into the published GISAID dataset, information for samples since April 2021 is less represented in the dataset.

7. The results on internal movements in figure 6 is very cool. Are there similar increases in importations following the relaxation of NPIs in February? (a lack of an increase might be caused by limited cases if they were falling during this time period.)

We thank the reviewer for the positive feedback, which helped us to make our work more clear. In response to the reviewer's question, we added this paragraph to the **Supplementary Results** section of supplementary materials (Lines 214-222):

“Increasing lineage importations into states after relaxation of the NPIs

After the relaxation of NPIs in February we also noticed an uptick in the number of importations into Germany, particularly evident in mid-March, with the rise in inter-state transmission becoming noticeable in mid-February (Fig. 6, S19). However, it is worth noting that the increase in the number of importation events into Germany appears to be slower. This could potentially be attributed to the impact of two border control NPIs, N9[variant travelers ban] (implemented on 2021-01-30) and N10[border closure] (implemented on 2021-02-14), which have a greater effect on importations into the country than inter-state importations, and were applied earlier in the timeline, influencing the observed pattern.”

We also added the following reference to this figure in the **Results** section of the main text (Lines 339-345):

“The inter-state spread of B.1.1.7 lineages started to rise before Christmas and reached a peak after the Christmas holidays (Fig. 6B), suggesting the impact of these holidays on the dissemination of these lineages within the country. Another peak occurred in late February, coinciding with the relaxation of internal restrictions, such as the closure of workplaces, movement restrictions to 15 kilometers, and the closure of schools. This suggests that the restrictions imposed by these internal NPIs had previously limited the spread of lineages across states. These two results were consistent in all three subsampling strategies (Fig. S18, S19).”

8. *The role of importations and internal migrations could be better placed in context of wave 3. For instance, how do these events and their timings correspond to cases counts throughout the study period? It could also be interesting to see how the rate of detecting new lineages (and the rate of lineage extinction) changes overtime and in response to NPI as well as the arrival of Alpha.*

*Thank you for this interesting suggestion. We have added a curve for the number of confirmed cases in Germany to the supplementary materials and a paragraph to the **results section** (Lines 318-326):*

*“Owing to effective NPIs, case numbers declined from November 2020 onwards, reaching a minimum in early February of 2021 (**Fig. S15, S16, S17**). However, this was followed again by a steep rise over the next two months. Already in January, most imported lineages were from the substantially more transmissible B.1.1.7 variant [32], as B.1.1.7 lineages propagated within the country and gained momentum, demonstrating reduced effectiveness of implemented NPIs for this more transmissible lineage or reduced effectiveness of remaining NPIs after relaxation of NPIs in February. This is consistent with the fact that B.1.1.7 was first detected in the UK in September 2020, and became the dominant lineage after a few months (in January it grew rapidly and continued to grow for three more months) [13,32].”*

Fig. S15 - Number of confirmed cases vs number of importation lineages and inter-state spreads. A) The weekly number of confirmed cases shown as a red line (with left axis scale) and the weekly number of importation lineages into Germany shown as bars with (with the right axis scale). B) The weekly number of confirmed cases (left axis scale) and the number of inter-state spread of lineages is shown as bars (with right axis scale).

Fig. S17 - Number of circulating lineages and its variation over time. A) Total number of circulating lineages over time, **B)** Changes in the number of circulating lineages over time. A 'circulating' lineage refers to a lineage whose presence, spanning from its TMRCA to its last sample, coincides with the study duration, represented by a week on the x-axis. The change in the number of circulating lineages (**B**) shows derivative of the number of circulating lineages (**A**), thus a positive (negative) value in the chart (**B**) shows an increase (decrease) in the number of circulating lineages.

Very Minor Concerns:

Line 235: The date format here is ambiguous. It would be more clear to use the YYYY-MM-DD format as in Figure 2.

We agree with the reviewer and we addressed this, as suggested.

Reviewer #2

This manuscript is a nice application of phylogeography to a very relevant public health issue. I really appreciate this work as it can be used widely in guiding future public health responses, especially promoting the use of free rapid antigen tests and mask-wearing for preventing SARS-CoV-2 and other respiratory viruses. Overall, I think this is a good manuscript and needs very little changes. Great work, and I look forward to your final manuscript!

We sincerely appreciate the positive assessment and appreciation of our work and the thoughtful comments, which we have found to substantially improve our manuscript further.

Major comments

Methods

1. - why 9 days for SIF? Why day 5 for the peak?

*We thank the reviewer for this comment. In response, we clarified the definition of the smoothed importation frequency in the **Methods** section (Lines 141-149) to:*

“To assess the effect of each nonpharmaceutical intervention on lineage importations, we defined the smoothed importation frequency (SIF) as follows (more details in the Supplementary Methods, Definition of the effectiveness measures): Let s be a smoothing parameter (to be used later), and set it to 5. Then, the SIF for each day is defined as a weighted sum of the number of importation events that happened on the current day and the next $2s-2$ days. The weight coefficients $w(j)$ are defined as $w(j) = j$ for $1 \leq j \leq s$ and $w(j) = 2s-j$ for $s+1 \leq j \leq 2s-1$, creating a triangular graph with a peak at $w(s)$. The measure is calculated as the sum of the number of importation events on the j -th day multiplied by the weight $w(j)$, which produces a smooth measure with a peak that focuses around the s -th following day.”

Second, we added the formula for the smoothed importation frequency and daily effectiveness measure to the supplementary materials (Lines 133-143):

“Definition of the effectiveness measures

Let s be the smoothing parameter and $\mu(i)$ be the importation frequency for day i . Then the measures SIF $\sigma(i)$, change rate of SIF $\sigma'(i)$, and effectiveness $\eta(i)$ on day i are calculated as follows:

$$\sigma(i) = \sum_{j=1}^s j \mu(i+j-1) + \sum_{j=s+1}^{2s-1} (2s-j) \mu(i+j-1)$$

$$\sigma'(i) = \sigma(i) - \sigma(i-1)$$

$$\eta(i) = \max_{t \in \{-6, \dots, 6\}} \left(\sigma'(i+t) - \min_{k \in \{\max(0,t), \dots, 6\}} \sigma'(i+k) \right)$$

Based on these formulas, $\eta(i)$ depends on $\sigma'(i-6), \dots, \sigma'(i+6)$, which depend on $\sigma(i-7), \dots, \sigma(i+6)$, which depend on the number of importations in $2s+12$ consecutive days: $\mu(i-7), \dots, \mu(i+2s+4)$. Here, the smoothing parameter is $s=5$, and the effectiveness depends on the number of importations in 22 consecutive days.”

Third, to further clarify the reasoning for the definition of the smoothed importation frequency and selection of value 5 for the smoothing parameter s , we added this paragraph to the supplementary materials (Lines 144-154):

“We aimed to limit the effectiveness assessment to a local scope of at most three weeks, to eliminate seasonal effects (as shown in (Gross et al. 2020) in 20 days a hard lockdown stabilizes the number of confirmed cases). By selecting a smoothing parameter (the parameter “s” in the definition of the effectiveness measures) not greater than five days, importations from more than three weeks later do not affect the calculated effectiveness of an NPI. Furthermore, to distinguish the effect of different NPIs, we maintained the smoothing parameter to be less than the minimum distance of six days between two consecutive NPIs. Therefore, we opted for the maximum possible value of five for the smoothing parameter to obtain a well-smoothed curve. In addition, when an NPI is implemented, it takes a few days for its effects to be observed between different states and cities. Therefore, we designed the triangular weighting function, to increment slightly every day from the actual day, and then decrease.”

Finally, to investigate the sensitivity of the SIF and the final results to the smoothing parameter, we performed further experiments and added these results to the supplementary materials (Lines 166-173):

“Sensitivity analysis of the smoothed importation frequency and effectiveness measures

To investigate the sensitivity of the final results to the smoothing parameter, we assessed the SIF measure for various values for the smoothing parameter. Increasing the smoothing parameter results in a smoother curve, while decreasing it reveals sharper changes in the curve. The actual SIF values also increase with higher smoothing parameters, for which a normalized chart is also provided. Notable peaks associated with effective NPIs, such as N8[mask] and N11[free test], are still discernible in all the curves.”

Fig. S12 - SIF for different values of the smoothing parameter. A) SIF curves calculated for different smoothing parameters. **B)** SIF curves normalized by the smoothing parameter squared. While increasing the smoothing parameter leads to higher absolute values of the SIFs, the normalized values maintain a relatively consistent average across different smoothing parameters. In this study, a smoothing parameter of 5 is employed.

We added the following sentence to the main text (Lines 307-310):

“Analysis of the sensitivity of the SIF and effectiveness measures show the robustness of the results with respect to minor changes in the smoothing parameter (Supplementary Materials, Sensitivity analysis of the smoothed importation frequency and effectiveness measures, Fig. S12).”

2. - Figure 2A. Is this for the total time period?

We thank the reviewer for this comment, which led us to clarify the caption of Figure 2A:

Fig. 2. Epidemiological properties of viral importation lineages. (A) Importation lineages, colored by the corresponding Pango-lineage to which they belong. The size of squares represents the size of the importation lineages. The figure exclusively displays lineages that were not eliminated before 2021-01-01.

3. - line 169/170: I apologise but I do not understand what you mean by splitting lineages, or what you are referring to when you say truly circulating lineages. Could you please define further?

We thank the reviewer for this comment. To clarify, we revised the paragraph to the following (now lines 182-186):

“This suggests that the inclusion of more non-German sequences results in the partitioning of samples of an inferred lineage into multiple lineages, thereby resulting in later TMRCA and smaller importation lineages. Consequently, since our sampling of non-German sequences was more limited than of German sequences, the number of lineages genuinely imported into Germany are likely to be higher than the obtained estimates.”

4. - was there any difference between states in Germany for sequencing proportions?

We thank the reviewer for the comment, which led us to clarify the passages in the manuscript relating to this question. Yes, the states in Germany have different sequencing rates through time. We added this table and figure describing sequencing rates for states to supplementary material and reference these now in the respective passage in the **results** (Lines 210-213):

“Although sequencing rates varied for states (Pearson CC of sequencing rate and population = -0.31, **Fig. S5**), there was no correlation with state-wise sequencing rates of the representative SARS-CoV-2 sequencing programme initiated for Germany in January 2021 (Pearson CC = -0.01, **Table S4, Fig. S6**).”

Table S4 - Sequencing rate for the state of Germany between 2020-09-25 and 2021-06-04.			
State	Number of confirmed cases	Number of sequences	Sequencing rate
Baden-Württemberg	446,331	11,286	2.53%
Bavaria	574,914	11,778	2.05%
Berlin	165,155	4,490	2.72%
Brandenburg	104,297	1,332	1.28%
Bremen	24,939	2,279	9.14%
Hamburg	68,992	4,161	6.03%
Hesse	269,780	2,286	0.85%
Lower Saxony	239,585	3,100	1.29%
Mecklenburg-Western Pomerania	42,822	1,210	2.83%
North Rhine-Westphalia	741,693	28,797	3.88%
Rhineland-Palatinate	143,218	5,725	4.00%
Saarland	37,725	1,617	4.29%
Saxony	277,569	10,427	3.76%
Saxony-Anhalt	96,254	3,498	3.63%
Schleswig-Holstein	58,723	794	1.35%
Thuringia	124,019	3,763	3.03%
Total	3,416,016	96,543	2.83%

Fig. S6 - Sequencing rate for the states of Germany over the course of the third wave. Lines reflect the seven-day moving average of the ratio of the number of sequenced genomes to confirmed cases for each state and Germany overall. Germany state abbreviations: BW: Baden-Württemberg, BY: Bavaria, BE: Berlin, BB: Brandenburg, HB: Bremen, HH: Hamburg, HE: Hesse, NI: Lower Saxony, MV: Mecklenburg-Western Pomerania, NW: North Rhine-Westphalia, RP: Rhineland-Palatinate, SL: Saarland, SN: Saxony, ST: Saxony-Anhalt, SH: Schleswig-Holstein, TH: Thuringia.

5. - When you are discussing the airports in Line 196 paragraph, I would also suggest making a clear point that with airport and air travel hubs and respiratory viruses, unless the airport is in a densely populated city, they are unlikely to be the area where the viruses is seeded due to the infection delay. Although many people may be infected during air travel, they are more likely to infect others at their final destination. To pull those dynamics apart though, you would need travel history metadata.

We thank the reviewer for this suggestion, which allows us to adapt the respective paragraph in the manuscript to the following (Lines 214-221):

“Notably, despite having the airport with the most annual number of travelers in Germany, comparatively few (<2%) importation lineages were first observed in Hesse. Among the ten most frequented airports in Germany in 2020, three are located in the state of North Rhine-Westphalia (ranked third, eighth, and tenth), one in Bavaria (ranked second), one in Baden-Württemberg (ranked seventh), while none is found in Mecklenburg-Vorpommern and Brandenburg [30]. While we anticipated that air travel would have a

substantial impact on lineage importations, we observed that the relationship to the state-wise population was more pronounced, indicating that the virus is more likely transmitted at their final destination.”

6 - Could you please include a table describing the NPIs you refer to in text! I think it is a really important thing to be able to see as you are reading the results.

We are grateful from the reviewer for this suggestion, which led to clarify figure 4 about NPI descriptions. We added a table to the end of Fig 4 containing the list of NPIs and their descriptions:

Fig. 4. Application date and descriptions for twelve major NPIs implemented in Germany between October 2020 and March 2021. The first seven rows show the details of internal measures over time, while the next row represents the border control measure. In the pink and blue rows, lighter colors represent less restrictive measures, and darker colors represent more restrictive interventions. Gray color indicates relaxation of the restrictions applied at the end of the third wave. Note that the presented intensity and colors are for the illustrative purposes only. N1[gathering restriction], N2[partial lockdown], N4[contact restriction], N5[lockdown], N8[mask], and N12[air travelers test] are internal NPIs, while N3[travelers registration], N6[UK travelers test], N9[variant travelers ban], N10[border closure], and N12[air travelers test] are border control NPIs. N7[15km movement ban] includes both internal and border control measures. In the table, the description of NPIs are presented. The pink rows represent internal NPIs, blue rows represent border control NPIs, and the purple row represents the NPI, which includes both internal and border control measures.

7 - I understand you are communicating many things in Figure 4, but I found it too confusing and took too long to interpret. I don't have any easy-fix recommendations for how to make it a better figure (apologies!), but at present, I don't think it is communicating what you need to the reader. Perhaps if you added a legend for the colours, and condensed some of the symbols? I also think the NPIs need to be written out or at least have some kind of description here and in Figure 5B.

We are thankful to the reviewer for this suggestion. To clarify, we extended the legend of Fig 4 to describe the colors: In the first figure, pink is for internal measures and blue is for border control measures. Lighter color (light pink for internal measures, light blue for border control) means less restriction in a measure, and darker color (dark pink for internal measures, dark blue for border control) means more restriction on a measure. Grey color means that restriction in a measure is relaxed. Please find Fig 4 in comment number 6.

We also added a very short description of the NPIs to the NPI labels in Fig 5:

Discussion

8 - When you discuss lockdowns, could you please define exactly what lockdown means in this context, as readers from different countries will have their own interpretations of what a lockdown is (in terms of intensity, NPIs, period).

We thank the reviewer for this suggestion. We have clarified this point, as suggested, in the main text as (Lines 358-362):

“Studies have shown declining lineage importations and persistence after lockdowns for the UK (commencing on 2020-03-23, involving the closure of non-essential shops and

services, and a stay-at-home order) and Switzerland (from 2020-03-17 to 2020-04-27, entailing the closure of bars, shops, and other gathering places except for essential services) [5,11], ...”

We have also incorporated the following changes into the introduction, including additional information about the lockdowns in the UK and Portugal (Lines 65-72):

“Both for the UK and Portugal, most introductions occurred prior to lockdown measures (UK lockdown on 2020-03-23, involving the closure of non-essential shops and services, and a stay-at-home order; Portugal lockdown on 2020-04-09 restricting people's movements between municipalities, closing air travel, and hardening border control), with the earliest ones becoming the largest and most persistent lineages post-lockdown [5]. Although Portugal quickly implemented lockdown measures, SARS-CoV-2 was likely circulating in late February, weeks before the first detected case [6].”

9 - When you discuss flu (line 308), could you maybe discuss further (i.e, what control measures are used to prevent flu, how could COVID be different?)

To address this, we have added the following paragraph to the discussion (Lines 382-395):

“Interestingly, studies of the dissemination of seasonal influenza viruses have shown an opposite effect, in that winter holidays delay epidemic peaks [38–40], likely because children play a key role in household transmission, due to their reduced immune protection compared to adults, which is a notable difference between seasonal epidemics and a pandemic involving novel infectious agent, where initial population-wide immune protection is low. Besides vaccination, key NPIs recommended for influenza by the World Health Organization (WHO) include practicing good hygiene, such as handwashing, covering the mouth and nose when coughing or sneezing, and avoiding close contacts when feeling unwell [41]. Examining the effectiveness of NPIs implemented to prevent SARS-CoV-2 on influenza, the three measures with highest observed decline in importations were gathering limitations or mask-wearing (applied together), travel bans or total border closures, and the closure of certain school levels [42]. This observation aligns with the understanding that influenza often spreads within schools, given lower immunity levels among children, in contrast to a pandemic, where a larger portion of the population lacks immunity.”

10 - you could probably discuss the lack of association of importations with airports in line 313/314

We addressed it as explained in the response to the comment 5.

11 - I would like to see you really highlight how effective non-invasive NPIs appear to be in controlling/preventing SARS-CoV-2! The travel bans that have been implemented in various countries have had harmful effects, both socially, economically and culturally. Yet, unfortunately, it will probably keep happening - but studies like this really show how we can reduce infectious disease better with simple measures.

Yes, indeed. We thank the reviewer for this suggestion to put more emphasis on an important aspect of the results. We extended the discussion to emphasize this point with the following paragraph (Lines 396-407):

“While our study demonstrates the overall effectiveness of various NPIs in controlling the importation of SARS-CoV-2 into Germany, it is crucial to acknowledge the potential negative socioeconomic impacts associated with certain measures, such as the closure of workplaces and travel bans. Furthermore, since elimination of a pathogen like SARS-CoV-2 is not feasible in countries like Germany, factors beyond the importation of new lineages also play an important role for pandemic preparedness. In light of these

considerations, our findings particularly highlight the efficacy of alternative interventions, specifically the implementation of freely available rapid testing (once available, since it needs to be developed for emerging pathogens) and the mandatory use of surgical or FFP2 masks in public places—two of the three most effective NPIs. These measures have proven to be both less harmful and highly effective in pandemic control, offering authorities a valuable and nuanced set of tools to navigate the challenging landscape of infectious disease management.”

Minor comments

Abstract:

- I think that because mask-wearing was such an important role in effective NPI use, you should highlight that in your second sentence, perhaps in between "antigenic testing, or travel restrictions". I.e.: "antigenic testing, physical transmission barriers (e.g. medical/surgical masks) or travel restrictions".

We thank the reviewer for this suggestion. We addressed this suggestion (Lines 22-26):

“Prior to the widespread availability of vaccines, nonpharmaceutical interventions such as reducing contacts, antigenic testing, physical transmission barriers (e.g. medical/surgical masks), or travel restrictions were the primary means of reducing viral transmission and case numbers, and quantifying the success of these measures is therefore key for future pandemic preparedness.”

- there are two commas that are not needed, in line 25 and 36, respectively between third, pandemic and few, negative.

We addressed this, as suggested.

Introduction

- a few minor grammatical changes I would recommend include line 57, changing it to: pandemic [4]. For example, previous studies suggest that it was transported from Hubei...

We addressed this, as suggested.

- Define UK as United Kingdom at first use, then change all U.K. to UK

We addressed this, as suggested.

- Line 85: I think that using "first emerged" is slightly controversial in terms of SARS-CoV-2, and I prefer using "first detected" as it is more accurate.

We have addressed this, as suggested (now Lines 90-91):

“The third infection wave in Europe during spring 2021 largely consisted of the B.1.1.7 (Alpha) lineage, which was first detected in Kent or Greater London [12].”

- You have defined NPIs but still define it a few times in the manuscript, where you could use the acronym, eg. line 91.

Thank you for pointing this out. We addressed this, as suggested.

Methods

- Could you please define the dates for the Christmas holidays in text and in Figure 2.

We have addressed this as suggested, in the caption of Figure 2B:

“The light orange block indicates the time of the Christmas holidays (2020-12-24 to 2021-01-09). The holidays vary by state, starting on 2020-12-24, and ending between 2021-01-02 and 2021-01-09.”

- Just a suggestion - I would use constantly or invariably instead of "Stably"

As the sentence shows a fact observable for all the three subsampling methods, we changed as follows (now lines 207-208):

“Consistently across all sampling methods, the fewest lineages first appeared in Mecklenburg-Western Pomerania and Brandenburg, both 0.5% (**Fig. S4**).”

Discussion

- Line 342 is a massive sentence - could be split into 2-3 smaller sentences.

We have addressed this, as suggested by changing the corresponding part to the following (now lines 433-439):

“Greater variation was observed for other properties, such as the TMRCAs and the absolute number of importation lineages. There was a tendency for an increase in the ratio of in-country to out-of-country samples and the absolute number of samples from within the country to lead to the merging of inferred importation lineages. This resulted in reduced absolute numbers of importation lineages and larger lineages with earlier TMRCAs. These findings suggest that caution is required when interpreting these properties.”

Reviewer #3

1. This study attempts to quantify the effectiveness of different non-pharmaceutical interventions (NPIs) on the rate and distribution of SARS-CoV-2 lineage importation events. This is an enormously difficult problem that is limited in part by the challenge of adjusting for variation in sampling rates over time and among regions not only within Germany (the focal country), but also sampling in other countries from which importation events must be inferred.

The authors employ a recently developed Bayesian method (Thorney BEAST) to generate a random sample of trees from the posterior distribution. This is an important feature of the analysis, but the main text of the manuscript lacks any explanation of how this method works, and there is limited mention in the Supplement.

We thank the reviewer for this important comment, which we addressed by providing a more extensive description of Thorney BEAST in the **Methods** section (Lines 121-127):

“To infer a time-calibrated phylogenetic tree for each dataset, we performed a Bayesian analysis with Thorney BEAST version 0.1.1 (https://beast.community/thorney_beast). As input for the Thorney BEAST analysis we used a template tree obtained from shrinking short branches, an initial inferred maximum likelihood tree as well as the dates of the sample nodes. The Thorney BEAST analysis infers internal node heights as well as resolutions of polytomies and produces samples from the posterior distribution of the fully resolved phylogenies.”

We furthermore extended the corresponding paragraph in the *Phylogenetic analysis* section of the supplementary materials (Lines 79-89):

“To infer a time-calibrated phylogenetic tree for each dataset, we performed a Bayesian analysis with Thorney BEAST version 0.1.1 (https://beast.community/thorney_beast). This method takes a multifurcating template tree to constrain topologies, as well as the topologically consistent uncollapsed trees from the previous step as starting trees, together with the sequence sampling dates as input. As by du Plessis et al (du Plessis et al. 2021) a strict molecular clock model was used, with an initial clock rate of $7.5 \cdot 10^{-4}$ substitutions/site/year and a skygrid coalescent tree prior. Further parameters were set as by (du Plessis et al. 2021), as specified in their BEAST configuration files. The BEAST configuration template file used here is available in the GitHub repository of the project at

<https://github.com/hzi-bifo/covid-germany-mcmc/blob/master/analyses/phylogenetic-test-snake-2/data/X.fixedRootPrior.skygrid-template-thorney.xml>”

2. From my quick reading of the associated literature, this method appears to constrain the tree topology to an input tree, instead resampling internal node heights based on the tip dates and a clock model. This implies that importations are fixed features of the input template tree. I'm concerned that reconstructing importations have a substantial level of uncertainty. SARS-CoV-2 phylogenies are difficult to resolve, resulting in many polytomies and nodes with low support. I think it would be important not only to quantify this uncertainty, but also to propagate it forward to measuring the impacts of different NPIs. Instead, key quantities such as daily effective measures (DEMs) are being reported without any error (e.g., line 244). This also applies to reconstructing the spread of lineages between countries and states. If the authors are not going to propagate uncertainty at all, then what is the point of using Bayesian sampling in the first place? It would have been much faster to rescale the tree in time by maximum likelihood, wouldn't it?

We are grateful to the reviewer for this thoughtful comment and find it crucial to clarify that our method does not rigidly constrain the tree topology to a fixed template. The approach involves a multi-step process. Initially, we create a maximum likelihood tree to capture the preliminary structure. Subsequently, we address the issue of unreliable information, primarily short branches, by shrinking them. Following

this, the Bayesian component comes into play, sampling from various resolutions of the uncertain parts and branch lengths, thereby introducing a level of flexibility to the reconstruction process. The Bayesian method allows that the uncertainty in polytomies and branch lengths is still represented in our results. In addressing uncertainty, it's noteworthy that our analysis, conducted independently on three sets of samples using different sampling strategies, spans a broad spectrum of uncertainty. The consistency of our results across these variations underscores the robustness of our method.

Also we are grateful for this comment, which led us to include an additional uncertainty analysis into our study. We introduced the propagated uncertainty analysis covered through the posterior distribution of the two Bayesian steps (phylogeny reconstruction and DTA steps).

*We added this paragraph to the manuscript section **Results** (Lines 312-317):*

“The 95% Highest Posterior Density (HPD) for the SIF and effectivenesses were analyzed based on samples from the posterior distribution of the two Bayesian steps (**Fig. 5A, S14**). When considering effectiveness based on the average values (**Table S6**), N11[free test] exhibited the highest average effectiveness, similar to the MCC tree. The second-highest average effectiveness was observed for N7[15 km movement ban], which closely resembled N8[mask] (13 days), one of the three most effective NPIs based on the MCC tree.”

Also, this paragraph is added to the supplementary methods (Lines 223-234):

“Analyzing effectiveness based on posterior samples of the Bayesian method

The posterior samples from tree topologies over which the Bayesian DTA was performed were retained for calculating errors and the 95% highest posterior distribution (HPD) for the smoothed importation and effectiveness measures. We calculated the smoothed importation and effectiveness values for every day for each individual posterior sample and compared their daily averages with the corresponding values obtained from the inferred MCC tree. The general trend of average smoothed importation values and average effectiveness of posterior samples aligns with the corresponding values calculated based on the MCC tree, demonstrating the robustness of the obtained results. Furthermore, to calculate a confidence interval for the inferred inter-state spreads of lineages, we calculated the number of inter-state movements for each of the posterior samples of the 16 state DTA (instead of the MCC) and reported 95% HPD and average values per week (**Table S6, Fig. S14**).”

And lines 346-350:

“The analysis of posterior distribution of the 16 state DTA (explained in section Analyzing effectiveness based on posterior samples of the Bayesian method of Supplementary Materials) shows that the results obtained from the MCC of the 16-state DTA and the average of the number of inter-state transmissions obtained from the Bayesian posterior distribution are similar. Both results fall within the 95% highest posterior density (HPD) region (**Fig. S14F**).”

Fig. 5. Effectiveness of NPIs. (A) SIF over time. The gray area represents 95% HPD. **(B)** Change rate of SIF over time. Red vertical bars indicate the implementation time of the twelve major NPIs in the country, and their heights represent the **effectiveness** metric for that day. **(C)** NPI effectiveness for Germany overall, as well as for North Rhine-Westphalia, Bavaria, Baden-Württemberg, and Saxony.

Table S6 - Comparison of effectiveness of MCC tree and posterior samples. The effectiveness of NPIs was calculated using inferred lineages for each posterior sample of the phylogenies, and the average and standard deviation are presented in the 'Average Effectiveness of Posterior Sampled Phylogenies' column. The '95% HPD' column represents the range of the number of importations, encompassing the 95% highest probability density.

NPI	Date	Effectiveness for MCC tree	Average effectiveness of posterior sampled phylogenies	95% HPD
N1	2020-10-19	17	18.671 ±4.761102	10-28
N2	2020-11-02	11	9.569 ±3.400885	3-16
N3	2020-11-08	6	5.431 ±2.530304	1-10
N4	2020-11-30	14	16.581 ±4.23053	9-24
N5	2020-12-16	16	15.5965 ±4.335215	8-24
N6	2020-12-22	10	13.5905 ±4.088419	6-21
N7	2021-01-11	16	19.0865 ±6.363825	7-31
N8	2021-01-24	24	15.3225 ±6.055149	4-26
N9	2021-01-30	10	17.9125 ±7.340762	4-31
N10	2021-02-14	12	11.154 ±4.189518	3-19
N11	2021-03-13	31	19.592 ±4.692286	10-28
N12	2021-03-30	12	13.2805 ±3.633238	7-20

B)

Effectiveness of Bayesian posterior samples

Smoothed importation of Bayesian posterior samples

C)

Fig. S14 - NPI effectiveness calculated based on the sampled phylogenies from the posterior distribution. The gray area represents the 95% HPD, MCC represents the inferred effectiveness based on the MCC tree, the blue line shows the average of the effectiveness values for the posterior samples. **A)** Effectiveness values, **B)** Logarithm of the effectiveness values. **C)** Smoothed number of importations (SIF), **D)** The change rate in the number of importations, **E)** The number of importations calculated based on the sampled phylogenies from the posterior distribution. **F)** Inter-state spread of the lineages from the posterior distribution. For each sample from the posterior distribution of the 16-state DTA Bayesian model, the number of inter-state spreads is calculated. The weekly average is represented by the blue line, and the 95% highest posterior density (HPD) is shown in the gray area. The weekly number of inter-state spreads of the lineages based on the MCC of the 16-state DTA is depicted as a red line.

3. Determining the effect of different NPIs on the variation in the rates of NPI among regions and over time is an exceedingly difficult problem, for instance because of confounding between the effects of different interventions. Each NPI category corresponds to a time period in which multiple NPIs of varying mechanisms and distribution are in effect. Moreover, there are a limited number of changes in interventions over time which limit our ability to evaluate causal relationships between each intervention and the NPI rate as a geographically- and temporally-structured outcome. Nevertheless, I found the authors' approach to be feasible and a reasonable advance on similar work in the literature.

We thank the reviewer for the careful and positive assessment.

Specific comments

* p.4, line 84: "The third infection wave in Europe during spring 2021 [...]" The authors also talk about third and fourth waves in the preceding paragraph, which is confusing. Are they making a distinction between waves in east Asia and Europe? If so, it would be helpful for this to be spelt out clearly.

We addressed this suggestion by clarifying the time periods of individual waves for the corresponding countries in manuscript as follows (Lines 59-64):

"For example, studies suggest that it was transported from Hubei, China, to multiple European countries several times between mid-January and early February 2020, before the large outbreak in northern Italy. The first wave of infections was studied in the United Kingdom (UK, late winter and early spring 2020) and Portugal (fall 2020), both of which had high early sequencing rates and therefore allowed to characterize the importation and diversity of lineages in depth [5,6]."

(Lines 74-79)

"Following the lifting of the first lockdown measures (Jun 2020 in UK), the B.1.117 lineage spread in a second infection wave across Europe over the summer of 2020 [8,9], leading to a persistently high volume of cases, despite B.1.117 having no notable transmission advantage. Further, a study of the third and fourth infection waves in Hong Kong (Jul and Nov 2021) provides insight into the successes and challenges of an elimination strategy [10], as opposed to the mitigation approach adopted by many other countries."

(Lines 94-96)

"Studying both the third and fourth waves in England (spring and summer of 2021), characterized by Alpha and Delta variants, respectively, revealed that their growth was initially masked by falling case counts of more dominant lineages [16]."

(Lines 97-102)

"Systematically assessing the effects of different NPIs on viral lineage importations and spread is key for future pandemic preparedness, although currently, these are not entirely clear [17]. Here we systematically analyzed how the NPIs that were successively implemented within a country affected SARS-CoV-2 lineage importations. For this, we used data from representative genomic surveillance collected over the course of the third pandemic wave in Germany (late 2020 and early 2021), together with large-scale Bayesian phylogenetic analyses."

* p4, lines 103-104: "Following established approaches [...], we subsampled viral genome sequences [...]" There are actually more methodological details provided in the Results (lines 158-161) and Discussion (lines 326-334) sections of the manuscript than in this Methods section, which is not helpful. Why can't this material be collected into Methods?

We have addressed this, as suggested and moved further details of the method into the **Methods** section (Lines 172-176):

"To address geographic and temporal sampling biases, assess the stability of results, and identify consistent properties across different data subsets, we created three distinct datasets using previously employed sampling strategies [7,8,11], namely Case ratio

subsampling, 50:50 subsampling, 25:100 subsampling (Methods, Table S2, Data sampling schemes in Supplementary Methods).”

Also, we moved the short description of the subsampling strategies to the **Methods** section as follows (Lines 113-121):

“To assess the consistency of results across sampling schemes, we created two more datasets using different genome subsampling schemes for phylogeographic analyses [5,21]. While we sampled sequences proportional to the number of confirmed cases in each country for each week as our main subsampling strategy (called case ratio subsampling strategy); we sampled 100 sequences from Germany and 100 sequences from all other countries for each week (called 50:50 subsampling strategy); and finally, up to 25 sequences from each other country, if available, and 100 sequences from Germany for each week (called 25:100 subsampling strategy). From each of these datasets, we identified SARS-CoV-2 lineages imported into Germany, along with their estimated importation times.”

Also, the part in the **Discussion** is shortened as follows (Lines 421-425):

“To mitigate temporal and regional, e.g. across countries and states, variations in sequencing of infected patient samples, we used the three sampling strategies on these data, namely case ratio, 50:50, and 25:100 subsampling strategies. This allowed us to assess the potential effects of common sampling strategies on the results, such as e.g. the numbers of identified importation lineages, their sizes and importation times across data sets.”

* line 115, Shannon Index and lineage evenness are not described in the Supplement.

We have addressed this, as suggested with adding the following to the supplementary methods (Lines 155-163):

Definition of the Shannon index and evenness

We calculated the Shannon Index [22] as a measure for calculating the number of large lineages in a state. A high number of SI shows a high number of lineages with a high number of samples. The evenness measure [23] is a normalized version of SI with less sensitivity to the number of different values. These measures are calculated as follows

$$SI = - \sum_{l \in L} \frac{s(l)}{S} \log \left(\frac{s(l)}{S} \right)$$

$$Evenness = \frac{SI}{\log(|L|)}$$

Where $s(l)$ is the number of samples of lineage l (in the corresponding state), and S is the total number of samples (in the same state). The summation is over all the lineages L (in that state).”

* line 116, why 16 states? what are these character states being mapped to?

We thank the reviewer for this comment which led us to clarify the ambiguity in the text about states. We added the clarification in the manuscript (Lines 132-136):

“To infer the spread of SARS-CoV-2 importation lineages within Germany between the 16 German states, we applied a 16-state Bayesian Discrete Trait Analysis (DTA) method,

each DTA state corresponding to one state of Germany, to the phylogenies of the inferred importation lineages using BEAST (Fig. 1B).”

* line 117, please define "DTA" at first use.

We have addressed this, as suggested (Lines 132-136):

“To infer the spread of SARS-CoV-2 importation lineages within Germany between the 16 German states, we applied a 16-state Bayesian Discrete Trait Analysis (DTA) method, each DTA state corresponding to one state of Germany, to the phylogenies of the inferred importation lineages using BEAST (Fig. 1B).”

* line 120 and onward, it would be helpful to provide some rationale for this smoothing function, i.e., why does it peak at four days post reference time point? This sentence: "We determined the effectiveness of an NPI considering [...]" is difficult to parse and would be easier to understand with a mathematical formula. I'm having trouble arriving at a 22 day time interval. Given NPI at time 0, the previous seven days takes our time line out to -7 and the next seven days to +7. At time +7, the SIF is calculated from time points +7 to +15.

We thank the reviewer for this comment. In response, we clarified the definition of the smoothed importation frequency in the **methods** section to (Lines 141-149):

“To assess the effect of each nonpharmaceutical intervention on lineage importations, we defined the smoothed importation frequency (SIF) as follows (more details in the Supplementary Methods, Definition of the effectiveness measures): Let s be a smoothing parameter (to be used later), and set it to 5. Then, the SIF for each day is defined as a weighted sum of the number of importation events that happened on the current day and the next $2s-2$ days. The weight coefficients $w(j)$ are defined as $w(j) = j$ for $1 \leq j \leq s$ and $w(j) = 2s-j$ for $s+1 \leq j \leq 2s-1$, creating a triangular graph with a peak at $w(s)$. The measure is calculated as the sum of the number of importation events on the j -th day multiplied by the weight $w(j)$, which produces a smooth measure with a peak that focuses around the s -th following day.”

Second, we added the formula for the smoothed importation frequency and daily effectiveness measure to the supplementary materials (Lines 133-143):

“Definition of the effectiveness measures

Let s be the smoothing parameter and $\mu(i)$ be the importation frequency for day i . Then the measures SIF $\sigma(i)$, change rate of SIF $\sigma'(i)$, and effectiveness $\eta(i)$ on day i are calculated as follows:

$$\sigma(i) = \sum_{j=1}^s j \mu(i+j-1) + \sum_{j=s+1}^{2s-1} (2s-j) \mu(i+j-1)$$

$$\sigma'(i) = \sigma(i) - \sigma(i-1)$$

$$\eta(i) = \max_{t \in \{-6, \dots, 6\}} \left(\sigma'(i+t) - \min_{k \in \{\max(0,t), \dots, 6\}} \sigma'(i+k) \right)$$

Based on these formulas, $\eta(i)$ depends on $\sigma'(i-6), \dots, \sigma'(i+6)$, which depend on $\sigma(i-7), \dots, \sigma(i+6)$, which depend on the number of importations in $2s+12$ consecutive days: $\mu(i-7), \dots, \mu(i+2s+4)$. Here, the smoothing parameter is $s=5$, and the effectiveness depends on the number of importations in 22 consecutive days.”

Third, to further clarify the reasoning for the definition of the smoothed importation frequency and selection of value 5 for the smoothing parameter s , we added this paragraph to the supplementary materials (Lines 144-154):

“We aimed to limit the effectiveness assessment to a local scope of at most three weeks, to eliminate seasonal effects (as shown in (Gross et al. 2020) in 20 days a hard lockdown stabilizes the number of confirmed cases). By selecting a smoothing parameter (the parameter “ s ” in the definition of the effectiveness measures) not greater than five days, importations from more than three weeks later do not affect the calculated effectiveness of an NPI. Furthermore, to distinguish the effect of different NPIs, we maintained the smoothing parameter to be less than the minimum distance of six days between two consecutive NPIs. Therefore, we opted for the maximum possible value of five for the smoothing parameter to obtain a well-smoothed curve. In addition, when an NPI is implemented, it takes a few days for its effects to be observed between different states and cities. Therefore, we designed the triangular weighting function, to increment slightly every day from the actual day, and then decrease.”

Finally, to investigate the sensitivity of the SIF and the final results to the smoothing parameter, we performed further experiments and added these results to the supplementary materials (Lines 166-173):

“Sensitivity analysis of the smoothed importation frequency and effectiveness measures

To investigate the sensitivity of the final results to the smoothing parameter, we assessed the SIF measure for various values for the smoothing parameter. Increasing the smoothing parameter results in a smoother curve, while decreasing it reveals sharper changes in the curve. The actual SIF values also increase with higher smoothing parameters, for which a normalized chart is also provided. Notable peaks associated with effective NPIs, such as N8[mask] and N11[free test], are still discernible in all the curves.”

Fig. S12 - SIF for different values of the smoothing parameter. A) SIF curves calculated for different smoothing parameters. **B)** SIF curves normalized by the smoothing parameter squared. While increasing the smoothing parameter leads to higher absolute values of the SIFs, the normalized values maintain a relatively consistent average across different smoothing parameters. In this study, a smoothing parameter of 5 is employed.

We added the following sentence to the main text (Lines 307-310):

“Analysis of the sensitivity of the SIF and effectiveness measures show the robustness of the results with respect to minor changes in the smoothing parameter (Supplementary Materials, Sensitivity analysis of the smoothed importation frequency and effectiveness measures, **Fig. S12**).”

** Figure 1A does not render well in greyscale - this would be resolved in part by using open and closed circles/squares in addition to colour.*

We thank the reviewer for this suggestion, which led us to fix the figure to be readable in greyscale. This is the revised figure along with a grayscale version:

The grayscale version:

* line 150, "lineage[s]"

We have addressed this, as suggested (Lines 167-168).

"The genomic surveillance program allowed us to conduct a systematic analysis of viral lineages imported into the country during this period using large-scale phylogenetic analysis techniques."

* I find the use of "lineage" to refer to both importations and PANGO designations to be somewhat confusing, e.g., repeated use in first line of Figure 2 legend; also line 207, "total number of lineages" refers to what?

We thank the reviewer for this comment, which led us to clarify the manuscript. We addressed this comment as suggested (Lines 224-242):

"To assess the diversity of importation lineages within states, we calculated the Shannon Index (SI), as by Spellerberg et al. [22], and the importation lineage evenness for each state. The SI is influenced by the total number of imported lineages and their respective size distributions, where having many imported lineages of similar sizes results in larger SI values. Across samplings, the state-wise SI strongly correlates with the logarithm of the

number of analyzed sequences. In the case of evenness, the SI of a state is divided by the number of imported lineages present in that state [23], providing a direct reflection of the importation lineage size distribution. SIs were highest for the population-rich states of North Rhine-Westphalia, Bavaria, and Baden-Württemberg, and Saxony, as expected due to the large number of cases and importation lineages observed. The lowest SI was determined for Schleswig-Holstein (**Fig. 3A**), which exhibited a substantially lower SI value than several other states with fewer confirmed cases and circulated importation lineages, as the lineage size distribution is strongly skewed towards one dominant lineage that entered the state before the Christmas holidays (**Fig. S7**), indicating an exceptionally low connectivity to other regions. This result remains stable across samplings (**Fig. S8**). With the exception of Hamburg, Bremen, and Schleswig-Holstein, lower lineage numbers correlated with more equal lineage sizes, higher evenness, across states (**Fig. 3B, S8**). While these relationships are not unambiguously resolved [23], such effects can be explained by niche preemption, which posits that a highly diverse environment, i.e., one with many lineages, is more challenging for incoming lineages to invade.”

We also added the following text into the method section to clarify that by default, when referring to lineage, we mean the importation lineage and not the pango-lineage (Lines 137-139):

"When referring to an importation lineage, the term 'lineage' is used for brevity throughout, while Pangolin lineage assignments are always explicitly referred to as such without explicitly mentioning 'importation' for brevity."

We also updated the legend and caption of Figure 2 in the manuscript:

Fig. 2. Epidemiological properties of viral importation lineages. (A) Importation lineages, colored by the corresponding Pango-lineage to which they belong. The size of squares represents the size of the importation lineages. The figure exclusively displays lineages that were not eliminated before 2021-01-01. (B) Estimated importation time and geographic distribution over states for the six largest B.1.1.7 importation lineages. The colors represent the number of viral genomes belonging to the respective importation lineage for each individual state. The light orange block indicates the time period of the Christmas holidays. (C) Number of importation lineages entered into the country per week, colored by the state that they were first observed in, shown individually for the states with the most importation events (dark pink: North Rhine-Westphalia, pink: Bavaria, dark blue: Baden-Württemberg, blue: Saxony) and for the remaining states in gray. The light orange block denotes the Christmas holidays, from 2020-12-24 to 2021-01-02 until 2021-01-09, depending on state, and the red arrow indicates the peak in importation events after Christmas holidays. See Fig S2 for the results for other samplings. The map represents the percentage of detected importation lineages, first observed in individual states (white: fewer, pink: more). NW = North Rhine-Westphalia, BW = Baden-Württemberg, BY = Bavaria, SN = Saxony.

* line 193, what do you mean by "CC", correlation coefficient? Please define at first use.

We have addressed this, as suggested (Lines 209-210):

"(Pearson Correlation Coefficient (CC) = 0.82)"

* line 202-204, "We hypothesize that the state population is more reflective of the ultimate destination of these travels, while airports serve as intermediate stops." This seems like a bit of a trivial statement - most people immediately leave the airport upon arrival. I surmise that the observed trend is due to the amplification of newly arrived lineages in large susceptible populations.

The reviewer is correct. We revised this sentence to clarify that although one might anticipate the highest number of importations in states with large airports, as many passengers may have these as their final destinations, this is not the case. These airports often serve as intermediate stops for many passengers. We updated the paragraph as follows (now lines 214-223):

"Notably, despite having the airport with the most annual number of travelers in Germany, comparatively few (<2%) importation lineages were first observed in Hesse. Among the ten most frequented airports in Germany in 2020, three are located in the state of North Rhine-Westphalia (ranked third, eighth, and tenth), one in Bavaria (ranked second), one in Baden-Württemberg (ranked seventh), while none is found in Mecklenburg-Vorpommern and Brandenburg [30]. While we anticipated that air travel would have a substantial impact on lineage importations, we observed that the relationship to the state-wise population was more pronounced, indicating that the virus is more likely transmitted at their final destination. We hypothesize that the state population is more reflective of the ultimate destination of these travels, as large airports often serve only as intermediate stops for travelers."

* line 227-228, "we gathered information about 4,000 national and local NPIs and summarized them into 12 major NPIs by date" - but earlier you said that you examined 110 NPIs (line 133)!

We have responded to this comment by rephrasing the sentence to provide the details as follows (Lines 247-253):

*"..., we summarized information of more than 4,000 NPIs implemented in Germany from published sources [24–26,31] into 119 NPIs (**Table S1**), and then categorized the national and local NPIs into 12 major NPIs (**Fig. 4**), and then classified them as internal NPIs (N1[gathering restriction], N2[partial lockdown], N4[contact restriction], N5[lockdown], N8[mask], N11[free test]), border control NPIs (N3[travelers registration], N6[UK travelers test], N9[variant travelers ban], N10[border closure], N12[air travelers test]), and NPI N7[15km movement ban], which includes both internal and border control measures."*

* line 237, I find it difficult to keep track of the different major NPIs with the N1-N12 numbering system, but I can't think of a better alternative.

To address this and similar comments from other reviewers, we modified Fig. 4 to include a table describing all the NPIs (see the following figure). Additionally, we provided a brief name for each NPI in the table, highlighting its most important features, and incorporated this name into the text wherever an NPI is mentioned. The updated figure is as follows.

Fig. 4. Application date and descriptions for twelve major NPIs implemented in Germany between October 2020 and March 2021. In the figure, the first seven rows show the details of internal measures over time, while the next row represents the border control measure. In the pink and blue rows, lighter colors represent less restrictive measures, and darker colors represent more restrictive interventions. Gray color indicates relaxation of the restrictions applied at the end of the third wave. Note that the presented intensity and colors are for the illustrative purposes only. N1[gathering restriction], N2[partial lockdown], N4[contact restriction], N5[lockdown], N8[mask], and N12[air travelers test] are internal NPIs, while N3[travelers registration], N6[UK travelers test], N9[variant travelers ban], N10[border closure], and N12[air travelers test] are border control NPIs. N7[15 km movement ban] includes both internal and border control measures. In the table, the description of NPIs are presented. The pink rows represent internal NPIs, blue rows represent border control NPIs, and the purple row represents the NPI, which includes both internal and border control measures.

* line 348, "By integrating a phylogenetic and phylogeographic analysis [...]" This amalgamation seems awkward to me, because phylogeography is essentially a specialized application of phylogenetics. I guess the authors are trying to allude to a hierarchical aspect of their analysis, i.e., ancestral state reconstruction model applied to phylogenies?

The comment is addressed and the text changed to the following (Lines 440-443):

"By integrating phylogeographic and discrete trait analyzes to infer lineage importations with a longitudinal assessment of importation dynamics, we provide an analytical framework for exploring the correlations between environmental factors, such as holiday and nonpharmaceutical intervention at a population level to lineage dissemination."

* line 355-359, "However, it is conceivable that confounding variables that coincide in time with a measure might contribute to the observed effects [...]" There is almost surely some confounding given the complexity of this system. What would be the expected effect of some of the most likely and strongest confounders?

We addressed it with changed the corresponding sentence in the manuscript as follows (Lines 448-451):

"However, it is conceivable that confounding variables that coincide in time with a measure might contribute to the observed effects, such as behavioral changes resulting from increased concern or NPI-related fatigue in the population [44], and travel patterns and incidence rate in neighboring countries [11]."

* Supplement lines 12-13, "Low-quality samples, corresponding to sequences shorter than 28,000 bp or with more than 1000 ambiguous bases, were removed as well [...]" What was the distribution of the number of ambiguous bases? Please provide a histogram summarizing this distribution. Sequences with substantial numbers of ambiguous base calls may be problematic for resolving phylogenies.

Here is the histogram of ambiguous sites in the sequences over which phylogenies were inferred. We included the following chart in the supplementary materials. Note that this procedure is the one suggested by sarscov2phylo (<https://github.com/roblanf/sarscov2phylo>). du Plessis et al. utilized 5% as the maximum threshold for allowed Ns in a sequence, whereas we opted for a more conservative threshold of 1000, which equates to approximately 3% of the sequence length.

Fig. S21 - Distribution of the number of ambiguous sites in the sequences for which the maximum likelihood phylogenies (FastTree method) were constructed. The distribution is calculated after application of the filterings and after removing non-informative sites as described in the method section.

We referenced the figure in the supplementary methods (Lines 71-74):

“Then it removes sites with more than 50% gaps, after converting N's to gaps, as well as sites suggested by <https://github.com/W-L/ProblematicSites SARS-CoV2/> and filters sequences that are shorter than 28,000 bp or have more than 1000 ambiguities (**Fig. S21**).”

* Supplement lines 16-18, “Bayesian phylogeography is a technique to study [...] pathogen spread, allowing integration of genomic information with different types of metadata and epidemiological spread models.” I think this definition is placing a strong emphasis on using a demographic/epidemiological model as a tree prior, but I don't think that Bayesian phylogeography is not restricted to this approach.

We clarified this statement to the following (Lines 16-20):

“Bayesian phylogeography refers to a field of study within evolutionary biology and biogeography that utilizes Bayesian statistical methods to infer the spatial and temporal patterns of the historical movement and evolution of taxa. This approach combines information from genetic data, such as DNA sequences, with geographic data to reconstruct the historical dispersal and divergence events of populations.”

* Supplement lines 24-28, this subsampling and stratification of data to a (relatively) small number of PANGO lineages (12) is a really important aspect of the study. What proportion of samples was discarded by restricting the analysis to this set of PANGO lineages? Are “derived” lineages included in these samples, or are these “pure” lineage sets? For example, are you including any of Q.1 (B.1.1.7.1), or B.1.36.*? In addition, what PANGO

lineage assignments are you using? Are you using the classifications provided by the GISAID database, or re-classifying sequences de novo? Which classifier and version of lineage definitions are you using?

Based on the comment, we added following paragraph to the supplementary materials (Lines 28-34):

“The number of samples of the selected Pango-lineages, after filtering out invalid dates, is 1,067,284 out of 1,729,077 (61.7%). The number of samples from the selected Pango lineages within Germany is 108,089 out of 119,801 (90.2%). Furthermore, the analyses are conducted independently on samples from each Pango-lineage. Consequently, the inferred lineages are entirely contained within one of the selected Pango-lineages. We relied on the Pango lineage assignment provided by GISAID in the metadata file.”

Also note that, This represents the distribution of Pango lineage versions in the original metadata file (prior to filtering out invalid dates): version 2021-05-27: 1808428, version 2021-05-19: 4931, version 2021-04-28: 4, version 2021-04-23: 27, version 2021-04-21: 2, version 2021-04-14: 3, version 2020-05-19: 3. Thus, 99.7% of the Pango-lineages are assigned with Pango-lineage version 2021-05-27.

In the metadata downloaded from GISAID on 2021-06-02, none of the samples are labeled as B.1.1.7.1 or B.1.36..*

** Supplement line 29, please clarify what you mean by "invalid dates".*

We have clarified this statement to the following (Lines 25-28):

“Of the sequences assigned to these Pango lineages, we removed sequences with dates not in the valid format or containing incomplete information (only year or year and month) or with dates prior to the first observed case of their respective Pango lineage, as in (Rambaut et al. 2020).”

** Supplement lines 53-58, you've named the subsampling strategies here quite clearly, but it would be helpful to provide at least a brief explanation in the main text.*

We thank the reviewer for this comment which led us to clarify the text about sub-samplings. We addressed this comment as suggested (Lines 113-121):

“To assess the consistency of results across sampling schemes, we created two more datasets using different genome subsampling schemes for phylogeographic analyses [5,21]. While we sampled sequences proportional to the number of confirmed cases in each country for each week as our main subsampling strategy (called case ratio subsampling strategy); we sampled 100 sequences from Germany and 100 sequences from all other countries for each week (called 50:50 subsampling strategy); and finally, up to 25 sequences from each other country, if available, and 100 sequences from Germany for each week (called 25:100 subsampling strategy). From each of these datasets, we identified SARS-CoV-2 lineages imported into Germany, along with their estimated importation times.”

** Supplement lines 50-51, "These further datasets contain 2689/2691 sequences from Germany and 1932/25,107 non-German sequences, respectively." It's not clear how you are using the forward slash here. We have to assume that the second number in each pair represents the third sampling strategy (25:100).*

We have clarified this by changing the statement in the supplementary material (Lines 55-57):

“These additional datasets include 2,689 and 2,691 sequences from Germany and 1,932 and 25,107 non-German sequences in the 50:50 and 25:100 datasets, respectively.”

* Supplement lines 75-76, there are quite a few BEAST settings missing here. Are you using exactly the same substitution model and prior hyperparameters used in du Plessis et al.?

To address this comment, we extended the corresponding paragraph in the supplementary materials as follows (Lines 79-89):

“To infer a time-calibrated phylogenetic tree for each dataset, we performed a Bayesian analysis with Thorney BEAST version 0.1.1 (https://beast.community/thorney_beast). This method takes a multifurcating template tree to constrain topologies, as well as the topologically consistent uncollapsed trees from the previous step as starting trees, together with the sequence sampling dates as input. As by du Plessis et al (du Plessis et al. 2021) a strict molecular clock model was used, with an initial clock rate of $7.5 \cdot 10^{-4}$ substitutions/site/year and a skygrid coalescent tree prior. Further parameters were set as by (du Plessis et al. 2021), as specified in their BEAST configuration files. The BEAST configuration template file used here is available in the GitHub repository of the project at

<https://github.com/hzi-bifo/covid-germany-mcmc/blob/master/analyses/phylogenetic-test-snake-2/data/X.fixedRootPrior.skygrid-template-thorney.xml>”

* Supplement line 79, “Convergence was ensured using LogAnalyser [...]” Convergence of a chain sample to the posterior distribution is never ensured.

We thank the reviewer for this suggestion. We have rephrased to (Lines 92-94):

“The LogAnalyser tool was used to calculate the Effective Sample Size (ESS), ensuring that all parameters were estimated with a minimum ESS value of 200.”

* Supplement line 97, this would be a good spot to provide formulae for Shannon Index and lineage evenness.

We have addressed this comment, by providing the formula for Shannon Index and lineage evenness in section Definition of the Shannon index and evenness of the supplementary materials (Lines 160-163):

“

$$SI = - \sum_{l \in L} \frac{s(l)}{S} \log \left(\frac{s(l)}{S} \right)$$

$$Evenness = \frac{SI}{\log(|L|)}$$

Where $s(l)$ is the number of samples of lineage l (in the corresponding state), and S is the total number of samples (in the same state). The summation is over all the lineages L (in that state).”

* Supplement line 104, what are the 16 states?

We thank the reviewer for this comment. We addressed the comment, as follows (Lines 118-121):

“To infer the spread of SARS-CoV-2 importation lineages within Germany between the 16 German states, we applied a 16-state Bayesian Discrete Trait Analysis (DTA) method, with each DTA state corresponding to one state of Germany, to the phylogenies of the inferred importation lineages using BEAST.”

** Supplement line 85, can you provide a reference for the maximum clade credibility tree? The majority of readers will not be familiar with this concept.*

We addressed this comment, as suggested (Lines 99-102):

“Then a maximum clade credibility (MCC) tree was created with TreeAnnotator for each sub-tree, with a location assigned to each internal node. The maximum clade credibility (MCC) tree is a summary of the posterior distribution of trees (Drummond and Bouckaert 2015).”

Reviewers' Comments:

Reviewer #1:

Remarks to the Author:

Thank you to authors for a very thorough response. I have only one very minor concern, remaining.

Line 130 refers to shrinking short branches. This should likely read 'collapsing' as in the supplemental methods.

Reviewer #2:

Remarks to the Author:

The authors have thoroughly addressed all comments and the revised manuscript is much improved.

Reviewer #3:

Remarks to the Author:

I thank the authors for their diligence in revising this manuscript. The presentation of results and description of analysis methodology are substantially improved. The notation for NPIs is cumbersome, but nevertheless an improvement on the previous approach (e.g., "N1"). I have no additional comments about the manuscript. However, I noticed that multiple scripts in the source code repository (<https://github.com/hzi-bifo/covid-germany-mcmc/tree/master>) contain hard-coded file paths that are specific to the user's directory (i.e., "/home/hforoughmand"). In addition, the scripts would benefit from some additional inline comments. Neither of these should be considered required changes.

Reviewer #1

Thank you to authors for a very thorough response. I have only one very minor concern, remaining.

We thank the reviewer for their thorough review of the manuscript and constructive comments that helped to substantially improve the manuscript.

Line 130 refers to shrinking short branches. This should likely read 'collapsing' as in the supplemental methods.

Addressed, as suggested.

Reviewer #2

The authors have thoroughly addressed all comments and the revised manuscript is much improved.

We thank the reviewer for the careful assessment of our work and constructive feedback.

Reviewer #3

I thank the authors for their diligence in revising this manuscript. The presentation of results and description of analysis methodology are substantially improved. The notation for NPIs is cumbersome, but nevertheless an improvement on the previous approach (e.g., "N1"). I have no additional comments about the manuscript. However, I noticed that multiple scripts in the source code repository (<https://github.com/hzi-bifo/covid-germany-mcmc/tree/master>) contain hard-coded file paths that are specific to the user's directory (i.e., "/home/hforoughmand"). In addition, the scripts would benefit from some additional inline comments. Neither of these should be considered required changes.

We thank the reviewer for the careful review of our code base. As suggested, we removed all the hard-coded paths (31 files are modified). More details on how the scripts should be used are given in the README.md file of the repository, from which individual scripts can then be accessed.